# The nuclear receptor HNF4 drives a brush border gene program conserved across murine intestine, kidney, and embryonic yolk sac

Lei Chen [1,2✉], Shirley Luo[1], Abigail Dupre[1], Roshan P. Vasoya[1], Aditya Parthasarathy[1], Rohit Aita [1], Raj Malhotra[1], Joseph Hur[1], Natalie H. Toke[1], Eric Chiles[2], Min Yang[1], Weihuan Cao [1], Juan Flores[3], Christopher E. Ellison[1], Nan Gao[3], Amrik Sahota[1], Xiaoyang Su [2,4], Edward M. Bonder[3] & Michael P. Verzi [1,2,5✉]

The brush border is comprised of microvilli surface protrusions on the apical surface of epithelia. This specialized structure greatly increases absorptive surface area and plays crucial roles in human health. However, transcriptional regulatory networks controlling brush border genes are not fully understood. Here, we identify that hepatocyte nuclear factor 4 (HNF4) transcription factor is a conserved and important regulator of brush border gene program in multiple organs, such as intestine, kidney and yolk sac. Compromised brush border gene signatures and impaired transport were observed in these tissues upon HNF4 loss. By ChIP-seq, we find HNF4 binds and activates brush border genes in the intestine and kidney. H3K4me3 HiChIP-seq identifies that HNF4 loss results in impaired chromatin looping between enhancers and promoters at gene loci of brush border genes, and instead enhanced chromatin looping at gene loci of stress fiber genes in the intestine. This study provides comprehensive transcriptional regulatory mechanisms and a functional demonstration of a critical role for HNF4 in brush border gene regulation across multiple murine epithelial tissues.

[1] Department of Genetics, Human Genetics Institute of New Jersey, Rutgers University, Piscataway, NJ, USA. [2] Rutgers Cancer Institute of New Jersey, Rutgers University, New Brunswick, NJ, USA. [3] Department of Biological Sciences, Rutgers University, Newark, NJ, USA. [4] Department of Medicine, Rutgers-Robert Wood Johnson Medical School, New Brunswick, NJ, USA. [5] Rutgers Center for Lipid Research, New Jersey Institute for Food, Nutrition & Health, Rutgers University, New Brunswick, NJ, USA. ✉email: lchen@dls.rutgers.edu; verzi@biology.rutgers.edu

The brush border is a characteristic of epithelia functioning in transport and absorption. It is composed of tightly packed microvilli; an exquisite, dense, finger-like array of apical cell surface protrusions that dramatically expand surface area of the epithelium. Brush borders line the surface of simple cuboidal and simple columnar epithelium found in different organs such as the intestine, kidney, and yolk sac[1–3]. Structurally, microvilli are composed of actin filament bundles, situated on a terminal web (also known as actin rootlets)[1,2]. Microvilli actin filaments are bundled by Villin[4], Espin[5], and Fimbrin (also known as Plastin-1)[6], and then attached to the plasma membrane by class I myosin motor proteins[7]. Anchoring microvilli into the terminal web increases stability of the brush border[8]. When the terminal web contracts, the microvilli spread apart and increase absorptive surface area[9].

Brush border anomalies are reported in many human diseases, such as microvillus inclusion disease, Crohn's disease, celiac disease, congenital tufting enteropathy, and congenital sodium diarrhea[2]. Understanding the regulatory mechanisms controlling brush border physiology is of great importance to human health. Most current brush border studies focus on genes encoding brush border structural proteins or cell biological characteristics of brush border assembly and function. For example, loss of actin-bundling proteins such as Villin, Espin, and/or Plastin-1 does not impair microvilli or only results in mild brush border anomalies, but can increase sensitivity to induced colitis[8,10–12]. Interruption of Eps8, an actin remodeler, caused reduced intestinal microvillus length and compromised intestinal fat absorption[13]. Disruption of Ezrin, a membrane-cytoskeleton crosslinking protein, leads to a disorganized terminal actin web[14]. However, it remains unclear how the transcriptional regulatory networks of brush border genes are established and whether they are common across brush border-containing tissues. We previously found that HNF4 paralogs activate enterocyte genes and are required for stabilization of enterocyte identity[15] and maturation of the fetal intestine[16]. HNF4 paralogs also regulate fatty acid oxidation and are required for renewal of intestinal stem cells[17]. In this study, using genetics and epigenetics approaches, we explore the transcriptional regulatory mechanisms of the brush border gene expression program and find that HNF4 binds and activates brush border genes. The brush border is severely disrupted upon HNF4 loss, suggesting HNF4 is a key regulator of the brush border in tissues of organs as diverse as the intestine, kidney, and embryonic yolk sac.

## Results

### Profiling of accessible chromatin points to a cross-tissue role for HNF4 in activating the brush border gene expression program.

In enterocytes (Fig. 1a), digestive enzymes and transporters are trafficked and inserted into the apical brush border to facilitate terminal digestion and absorption of nutrients[18]. Similarly, the cuboidal epithelial cells of the renal proximal tubules (Fig. 1a) also produce a brush border facing the lumen of the tubule to facilitate the renal functions of reabsorption and secretion[19]. Interestingly, a brush border also lines cells of the murine yolk sac epithelium (Fig. 1a), which helps transport maternal nutrients during embryonic development[3,20]. We sought to investigate similarities and differences of brush border transcripts across these three tissues.

We examined the overlap of annotated brush border transcript expression between the intestine, kidney, and yolk sac (the brush border-related gene list, Supplementary Data 1, is composed of brush border genes (GO:0005903), brush border membrane genes (GO:0031526), and microvillus genes (GO:0005902) from the Gene Ontology knowledgebase[21,22]). We performed RNA-seq in

adult intestine and kidney, and curated public RNA-seq data of embryonic yolk sac[20], to find the vast majority of brush border genes are commonly expressed (FPKM (fragments per kilobase of transcript per million mapped reads) or RPKM (reads per kilobase of transcript per million mapped reads) > 1) across all 3 tissues (Fig. 1b and Supplementary Fig. 1a). Genes known to contribute to documented brush border characteristics such as F-actin bundling, membrane-cytoskeleton crosslinking, intermicro-villar adhesion genes[1], and membrane/transport functions were commonly expressed in all three tissues (Fig. 1c), indicating similar features between brush borders of different tissues.

Given the similarities in brush border transcriptomes between these tissues, we sought to identify whether brush border gene regulatory programs could be conserved between the tissues. We employed DNase I hypersensitive sites sequencing (DNase-seq) and RNA-seq to identify transcription factors operating at these accessible chromatin regions of brush border genes (see strategy in Fig. 1d). We first collected presumed regulatory regions within 50 kb of annotated brush border gene promoters using DNase-seq data of the intestine[23] and kidney[24] (Model-based Analysis of ChIP-Seq (MACS), $P \leq 10^{-5}$), and compared these regions between tissues. We found that intestine and kidney exhibited a strikingly similar chromatin accessibility at brush border genes (Figs. 1e, f). Subsequent mining of these regions for transcription factor-binding motifs revealed that the HNF4 DNA-binding site was the top transcription factor motifs enriched in these regions (Fig. 1g and Supplementary Data 2, the architectural protein, CTCF, motif was the top overall motif). Of the transcription factors with motifs enriched at these presumed brush border regulatory regions, HNF4A is highly expressed in all three brush border-containing tissues (Figs. 1h, i). Together, these epigenomic analyses suggest that HNF4 could be a fundamental regulator of brush border gene expression across tissues.

### HNF4A binds chromatin in a strikingly similar pattern in the intestine and kidney.

Analysis of our RNA-seq data indicated that 71% of brush border genes are commonly expressed in both the intestine and kidney of adult mice (Fig. 1b). For example, Espn (encodes actin-bundling protein) and Anks4b (functions in inter-microvillar adhesion and brush border assembly) are both expressed in the intestine and kidney (Supplementary Fig. 1a). In addition, there are 20 brush border genes only expressed in the intestine, whereas 33 brush border genes are only expressed in the kidney (Fig. 1b). Myo1a exemplifies a gene expressed in the intestine, whereas Myo1c is expressed in the kidney (Supplementary Fig. 1b, c). Besides these structure-based brush border genes, we also examined function-based brush border genes in these different tissues. We found Npc1l1, a cholesterol transporter, is only expressed in the intestine, whereas Slc22a12, a urate transporter, is only expressed in the kidney (Supplementary Fig. 1b, c).

We wondered whether brush border gene regulatory mechanisms were similar or divergent in these tissues and hypothesized that HNF4 could play a critical and conserved role in brush border gene regulation. We therefore compared HNF4A binding to the intestinal and renal genomes using chromatin immuno-precipitation sequencing (ChIP-seq) (Supplementary Fig. 2a). Strikingly similar HNF4A-binding profiles were observed between the intestine and kidney, with the vast majority of ChIP-seq-binding sites (90%) identified as common between the two tissues (11,017 sites, false discovery rate (FDR) < 0.01, DiffBind[25], Fig. 2a). DNA-binding sequence motif analysis of intestinal and renal HNF4A ChIP-seq binding regions revealed similar enrichment of transcription factor-binding motifs (Supplementary Fig. 2b and Supplementary Data 3). Quantitatively, relatively few HNF4A-binding regions were identified as

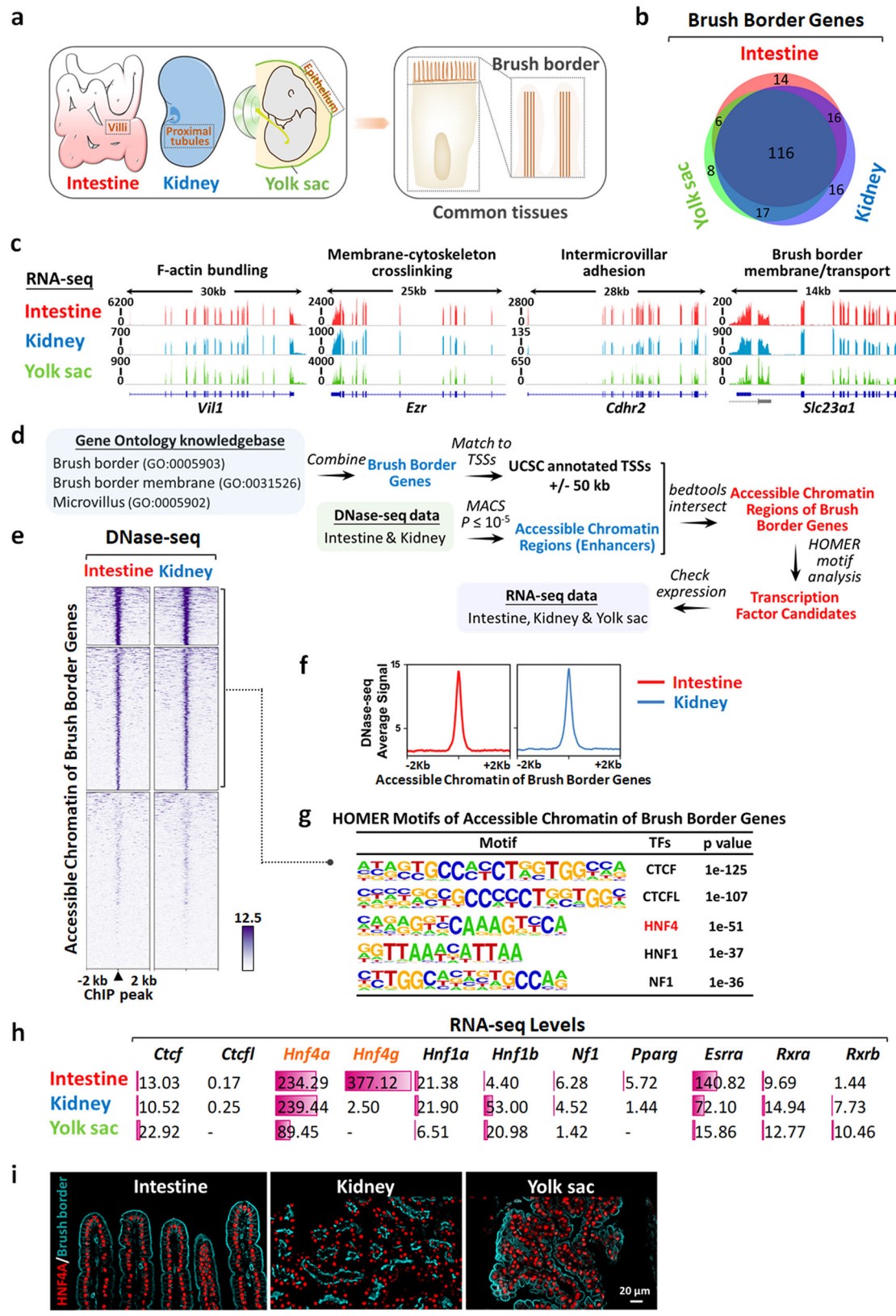

differentially bound in the intestine vs. kidney; differentially bound regions were referred to as intestine-enriched sites (732 sites, FDR < 0.01, DiffBind[25], Supplementary Fig. 2c and Supplementary Data 4) and kidney-enriched sites (473 sites, FDR < 0.01, DiffBind[25], Supplementary Fig. 2c and Supplementary Data 4). HNF4A-binding signals are more robust in the intestine

than in the kidney, in the regions that are identified as intestine-enriched sites. Similarly, HNF4A-binding signals are more robust in the kidney than in the intestine, in the regions that are identified as kidney-enriched sites (Supplementary Fig. 2d). Consistent with functional regulatory elements, tissue-enriched HNF4A-binding sites coincide with tissue-specific chromatin

**Fig. 1 HNF4 is the top transcription factor candidate for regulating brush border genes. a** The brush border is a common feature of the intestine, kidney, and yolk sac. **b** Venn diagram shows that most of the brush border genes are commonly expressed in the intestine (RNA-seq FPKM > 1), kidney (RNA-seq FPKM > 1), and yolk sac (RNA-seq RPKM > 1, PRJEB18767). **c** Examples of brush border gene expression in the intestine, kidney, and yolk sac. **d** Strategy to identify transcription factors controlling brush border genes. TSSs: transcription start sites. **e** Accessible chromatin regions of brush border genes (±50 kb of TSSs, enhancers) were profiled by DNase-seq data in the intestine and kidney (GSE57919 and GSE51336, $n = 2$). k-means = 3 was used to group regions with strong DNase-seq signals. **f** Average signal of DNase-seq in accessible enhancer chromatin regions of brush border genes ($n = 2$). **g** HOMER motif analysis of accessible chromatin regions of brush border genes (see full list in Supplementary Data 2, MACS $P \leq 10^{-5}$, enhancer sites). Statistical tests were embedded in the MACS and HOMER packages. **h** RNA transcript levels of top transcription factors identified from HOMER motif analysis in the intestine, kidney, and yolk sac (RNA-seq mean FPKM or RPKM values) indicate that HNF4 is a top candidate regulator of brush border genes. **i** Immunofluorescence co-staining of HNF4A and brush border marker ($n = 3$ biologically independent mice) in the intestine (HNF4A/β-actin), kidney (HNF4A/ABCG2), and yolk sac (HNF4A/β-actin).

accessibility (Supplementary Fig. 2e, f) and tissue-specific gene expression (Supplementary Fig. 2g). Genes linked to the intestine-enriched sites of HNF4A ChIP-seq are more highly expressed in the intestine than in the kidney, whereas genes associated with the kidney-enriched sites of HNF4A ChIP-seq are more highly expressed in the kidney than in the intestine (Supplementary Fig. 2g).

**HNF4 directly activates the majority of the brush border gene expression program.** In the intestine, HNF4A acts redundantly with HNF4G, an intestine-restricted HNF4 paralog, and these two paralogs activate enhancer chromatin and stabilize enterocyte cell identity[15]. Unlike HNF4G, HNF4A is expressed in both the intestine and kidney (Fig. 1h). The molecular consequences of HNF4 dependency in the intestine and kidney were appreciated through transcriptome analysis by a tamoxifen-inducible intestine-specific knockout *Villin-Cre^ERT2; Hnf4α^f/f; Hnf4γ^Crispr/Crispr* mouse model (hereafter referred to *Hnf4αγ^DKO*) and a tamoxifen-inducible knockout in the kidney using a *UBC-Cre^ERT2; Hnf4α^f/f* mouse model (hereafter referred to *Hnf4α^KO*), respectively. In all, 410 genes were significantly downregulated in both the intestine (*Villin-Cre^ERT2; Hnf4αγ^DKO*) and kidney (*UBC-Cre^ERT2; Hnf4α^KO*) knockouts of HNF4 ($\log_2$ fold-change < −1, FDR < 0.05, Fig. 2b), and these common downregulated genes were much more enriched for ontologies associated with brush border and transport functions than genes that were only downregulated in one of the two tissues (Fig. 2c and Supplementary Data 5). Gene set enrichment analysis (GSEA) confirmed that brush border genes were disproportionately downregulated in both the intestine and kidney upon HNF4 loss (Fig. 2d and Supplementary Fig. 3). To understand how HNF4 factors are impacting the transcriptome of these brush border genes, we next investigated the relationship between HNF4 chromatin-binding events and brush border gene expression. Seventy-eight brush border genes were significantly downregulated in RNA-seq analysis of the intestines of *Villin-Cre^ERT2; Hnf4αγ^DKO* (FDR < 0.05, Supplementary Fig. 3) and 50 of them are directly bound by HNF4 in the intestine via ChIP-seq (genes with underline, MACS $P \leq 10^{-3}$, Supplementary Fig. 3). Similarly, 85 brush border genes were significantly downregulated in the kidneys of *UBC-Cre^ERT2; Hnf4α^KO* (FDR < 0.05, Supplementary Fig. 3) and 49 of them are directly bound by HNF4 in the kidney (genes with underline, MACS $P \leq 10^{-5}$, Supplementary Fig. 3). For example, HNF4 can directly bind and activate *Myo7b* (involved in intermicrovillar adhesion) and *Espn* in both the intestine and kidney (Fig. 2e). These findings suggest that HNF4 factors bind to brush border gene chromatin and directly activate an entire catalog of brush border gene expression.

We next sought to more specifically associate HNF4 binding to active enhancer chromatin structure and chromatin looping, to target brush border genes. We first profiled enhancer chromatin structures of brush border genes in the intestinal epithelial cells of

control and *Hnf4αγ^DKO* using H3K27ac MNase ChIP-seq. Decreased H3K27ac signals (an active chromatin marker) were observed in *Hnf4αγ^DKO* compared to wild-type (WT) control (Supplementary Fig. 4a, b). The levels of the remaining H3K27ac signal were observed at the center of the regions in *Hnf4αγ^DKO*, suggesting more restricted chromatin accessibility of these brush border genes upon loss of HNF4 paralogs (Supplementary Fig. 4b). We also investigated whether HNF4 binding impacted chromatin looping between presumed enhancers and transcriptional promoters of brush border genes using H3K4me3 HiChIP to capture three-dimensional (3D) chromatin configurations at brush border gene loci. There are 152 brush border genes with FPKM > 1 in the intestinal epithelium and 44 of these 152 genes show disrupted looping events in *Hnf4αγ^DKO* (DEseq2 $P < 0.05$, Genomic Regions Enrichment of Annotations Tool [GREAT] 10 kb). Examples of genes critical to brush border structure and function illustrate the contribution of HNF4 transcription factors to chromatin looping at these brush border genes (Fig. 2f). In each example of brush border-related genes, including F-actin bundling genes (e.g., *Espn* and *Pls1*), membrane-cytoskeleton crosslinking genes (e.g., *Ezr* and *Myo1a*), intermicrovillar adhesion genes (e.g., *Myo7b* and *Cdhr5*), and brush border membrane/transport genes (e.g., *Slc6a19* and *Npc1l1*), HNF4 directly binds (HNF4 ChIP-seq tracks, Fig. 2g and Supplementary Fig. 4c) and is required for maintaining active chromatin (H3K27ac MNase ChIP-seq tracks, Fig. 2g and Supplementary Fig. 4c) and transcript levels (RNA-seq bar charts, Fig. 2g and Supplementary Fig. 4c, d). In each case, reduced chromatin looping events were observed upon HNF4 loss in the intestinal epithelium (differential loops of H3K4me3 HiChIP-seq, DEseq2 $P < 0.05$, Fig. 2g and Supplementary Fig. 4c). Taken together, multiple-omics approaches demonstrate a mechanism through which HNF4 binds to brush border gene loci, maintains active and accessible nucleosome structures, and promotes 3D chromatin looping between regulatory elements and brush border gene promoters to activate their transcription.

**HNF4 transcription factors are required for intestinal brush border.** The strong regulatory relationship between HNF4 and brush border gene regulation prompted us to characterize the consequences of HNF4 loss on brush border function in each of these brush border-containing tissues. We recently found impaired intestinal function upon HNF4 loss by using *Villin-Cre^ERT2* [15] and *Shh-Cre* [16] mouse models, and one of the top Gene Ontology categories of HNF4-dependent genes in the intestine is associated with brush border[15]. However, the extensive regulatory mechanisms of HNF4 transcription factors in the intestine and other brush border-containing tissues have not been dissected yet. Given the important roles of HNF4 in brush border gene expression, we further investigated whether HNF4 paralogs are required for intestinal brush border by a tamoxifen-inducible, intestine-specific knockout (Fig. 3a). HNF4A and HNF4G

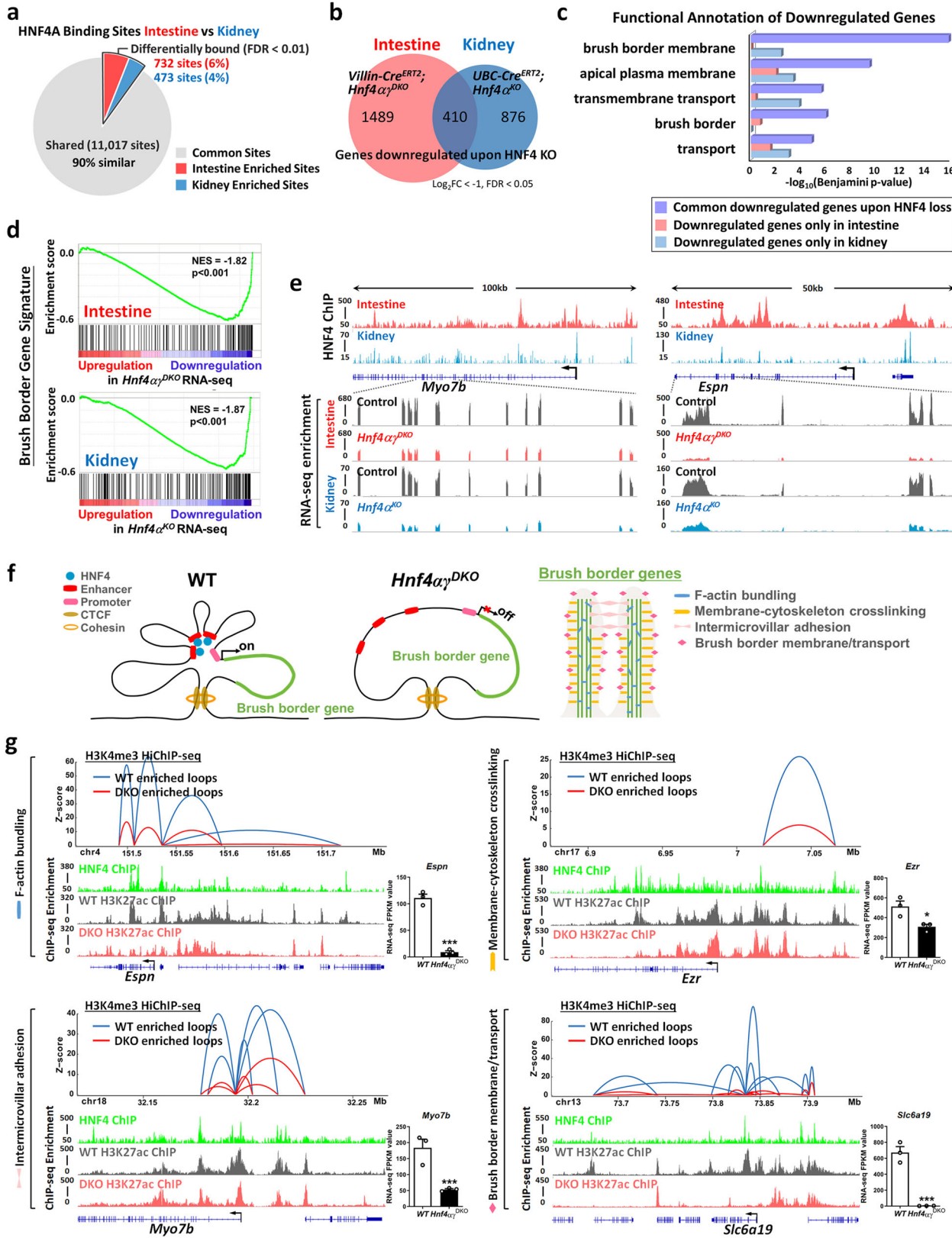

transcription factors are expressed throughout the intestinal epithelium (top panel, Supplementary Fig. 5a). Phalloidin and β-actin were used to mark the brush border at the apical surface of intestinal epithelial cells and was clearly found in WT, but was compromised in *Hnf4αγ*^DKO (Fig. 3b, c and Supplementary Fig. 5). Expression of brush border markers, such as alkaline

phosphatase (AP, a brush border membrane-bound enzyme), β-actin (localized to the apical cytoskeleton), Villin (actin-bundling protein of the brush border), and Ezrin (linking the cytoskeleton to the cell plasma membrane of the brush border) were all severely disrupted upon HNF4 loss in the intestine (Fig. 3d). In addition, the intestinal cells were nicely aligned in WT, but this

**Fig. 2 HNF4 transcription factors bind and activate brush border genes and are required for chromatin looping at brush border gene loci. a** ChIP-seq of HNF4A in the intestine and kidney, followed by DiffBind analysis (FDR < 0.01, GSE112946, and GSE47192, $n = 2$) shows that 11,017 sites are shared, and only 732 and 473 sites are differentially bound by HNF4A in the intestinal epithelium (MACS $P \leq 10^{-3}$) and kidney (MACS $P \leq 10^{-5}$), respectively. **b** Venn diagram shows the numbers of downregulated genes in the intestine (*Villin-Cre^ERT2; Hnf4αγ^DKO*, $n = 3$ biologically independent mice) and kidney (*UBC-Cre^ERT2; Hnf4α^KO*, $n = 4$ biologically independent mice) knockouts of HNF4 with log$_2$ fold-change < −1, FDR < 0.05, as evidenced by RNA-seq. **c** In HNF4 mutants, brush border-related functions are enriched in the common downregulated genes of the intestine and kidney (410 genes, purple bar) compared to genes only downregulated in *Hnf4αγ^DKO* intestine (1489 genes, red bar) or *Hnf4α^KO* kidney (876 genes, blue bar). Benjamini $P$-values were calculated using DAVID. **d** GSEA of RNA-seq data reveals that the brush border gene transcripts are compromised upon HNF4 loss in both intestine and kidney knockouts (Kolmogorov–Smirnov test, one-sided for positive and negative enrichment scores, $P < 0.001$). **e** Examples of HNF4 ChIP-seq ($n = 2$ in each tissue) and RNA-seq tracks (Intestine: $n = 3$; Kidney: $n = 4$) at brush border-related gene loci. **f** Working model of how HNF4 factors are impacting brush border genes. Brush border genes can be classified into F-actin bundling genes, membrane-cytoskeleton crosslinking genes, intermicrovillar adhesion genes, and brush border membrane/transport genes. HNF4 can bind and facilitate long-range chromatin interactions at genes from each category. **g** Examples of compromised chromatin looping and compromised active chromatin markers at loci of brush border genes. H3K4me3 HiChIP-seq was done in villus cells of *Hnf4αγ^DKO* and their littermate controls. Differential loops (DEseq2 $P < 0.05$) are visualized by Sushi package for loops with $q \leq 0.0001$ and counts $\geq 8$ (combined 2 replicates). Bar charts show transcript levels of brush border genes. Statistical tests were embedded in Cuffdiff. The data are presented as mean ± SEM ($n = 3$ biological replicates, ***Cuffdiff FDR < 0.001, and *FDR < 0.05). H3K4me3 HiChIP-seq (GSE148691): $n = 2$ biological replicates; H3K27ac ChIP-seq (WT vs. *Hnf4αγ^DKO*; GSE112946): $n = 2$ biological replicates; HNF4 ChIP-seq (GSE112946): $n = 2$ biological replicates; RNA-seq (WT vs *Hnf4αγ^DKO*; GSE112946): $n = 3$ biological replicates. More examples are shown in Supplementary Fig. 4c.

---

organized structure was completely lost upon HNF4 loss (Fig. 3b–d). In *Hnf4αγ^DKO*, protein levels of HNF4A were significantly diminished after 3 days of tamoxifen induction and brush border-related proteins (e.g., Villin, Keratin 20) were dramatically downregulated in a time-dependent response to HNF4 loss (Fig. 3e). The total protein levels of β-actin were not altered upon HNF4 loss (Fig. 3e), but the distribution of β-actin in the cells was disrupted (Fig. 3c, d). To more closely examine the consequence of HNF4 factor loss on the brush border, electron microscopy was used to visualize the ultrastructure of brush border. We found the expected finger-like projections of a "brush border" on the surfaces of enterocytes in WT (top panel, Fig. 3f), but enterocytes from *Hnf4αγ^DKO* exhibited significant alterations in the brush border architecture, with disorganized and shortened microvilli (Fig. 3f, h). Cross-sections through microvilli revealed organized, uniform, and tightly packed arrays in WT, but not in *Hnf4αγ^DKO* (Fig. 3g). Compared to WT controls, individual microvilli were larger in diameter and shorter in height in *Hnf4αγ^DKO* (Fig. 3f–h). Together, these data indicate that the brush border is severely disrupted upon HNF4 loss in the intestinal epithelium, suggesting that HNF4 factors are required for intestinal brush border.

**HNF4A is required for renal brush border.** Besides lining the intestinal epithelium, a brush border is also found in the proximal tubules of the kidney. Deletion of *Hnf4a* by different embryonic-onset Cre drivers in the nephrons leads to defects in the development of proximal tubules[26,27], but little is known about the adult-onset disruption of *Hnf4a* in the kidney and its impact on brush border. To study the function of HNF4 in the renal brush border (Fig. 4a), we took advantage of the fact that HNF4A is expressed in both the intestine and kidney, whereas HNF4G is not expressed in the kidney (Fig. 4b, c). We therefore created both developmental- and adult-onset knockout of *Hnf4a* in a tamoxifen-inducible *UBC-Cre^ERT2; Hnf4a^f/f* mouse model (hereafter referred to *Hnf4α^KO*), which deletes all kidney HNF4 without compromising intestinal function. By labeling renal proximal and distal tubules with ABCG2 (an ATP-binding cassette transporter that also functions as a uric acid transporter) and NCC (a sodium chloride cotransporter), respectively, we confirmed that HNF4A is only expressed in the proximal tubules within the kidney (Fig. 4f and Supplementary Fig. 6a), a finding confirmed by re-analysis of public single-cell RNA-seq data[28] (Supplementary Fig. 7). AP marks the brush border of proximal tubules and its activity gets stronger over developmental time,

from birth to adulthood (Supplementary Fig. 6b). AP activity was lost in the kidney of *Hnf4α^KO* but was not affected in *Hnf4γ^KO* (Fig. 4d), indicating that HNF4A is required for renal brush border but HNF4G is not required. AP and HNF4A are colocalized in the proximal tubules of the kidney (Supplementary Fig. 6c), and AP activity is positively correlated with the expression level of HNF4A (Fig. 4e), suggesting that HNF4A works cell-autonomously in the proximal tubule epithelium. In addition, HNF4 can directly bind and activate *Abcg2* in both the intestine and kidney (Supplementary Fig. 8a). ABCG2 is normally expressed in the brush border of both the intestine and kidney, and is dramatically compromised upon HNF4 loss in both tissues at the transcript and protein level (Supplementary Fig. 8a, b).

To broaden the scope of HNF4 gene regulation in the kidney, RNA-seq was conducted and found a robust shift in the the *Hnf4α^KO* kidney transcriptome away from proximal tubule transcripts and towards distal tubule transcripts (Fig. 4g, Supplementary Fig. 9a and Supplementary Data 6. Gene lists defined from ref. [28]). Adult-onset knockout of HNF4A also resulted in decreased expression levels of brush border markers (Supplementary Fig. 9b) and reduced kidney weight (Supplementary Fig. 9c). Electron micrographs confirmed that both adult (Fig. 4h) and developing kidneys of *Hnf4α^KO* (Supplementary Fig. 9d) exhibited diminished and sparse brush border microvilli in proximal tubules; distal tubules were unaffected upon HNF4A loss (Fig. 4h).

Proximal tubules play a prominent role in reabsorption of water, electrolytes, low-molecular-weight proteins, glucose, and amino acids. They also function in regulating acid–base balance by reabsorbing filtered bicarbonate. As an abnormal renal brush border can lead to malfunction of the proximal tubules, we collected urine and plasma for signs of renal malfunction. Compared to littermate controls, *Hnf4α^KO* showed decreased urine volume and pH but increased urine-specific gravity and protein (Fig. 4i), consistent with diminished proximal tubule function. Fanconi syndrome is a disorder of the kidney proximal tubule function and results in decreased reabsorption of certain substances into the bloodstream[29]. Coincidentally, increased concentration of glucose was observed in the urine of *Hnf4α^KO* (Fig. 4j), which is consistent with the glycosuria symptom of Fanconi syndrome. In addition, decreased concentrations of total carbon dioxide, potassium, calcium, and glucose were also observed in the plasma of *Hnf4α^KO* (Fig. 4k). Taken together, these data indicate that HNF4A loss results in compromised proximal tubule brush border and dysfunction of tubular reabsorption.

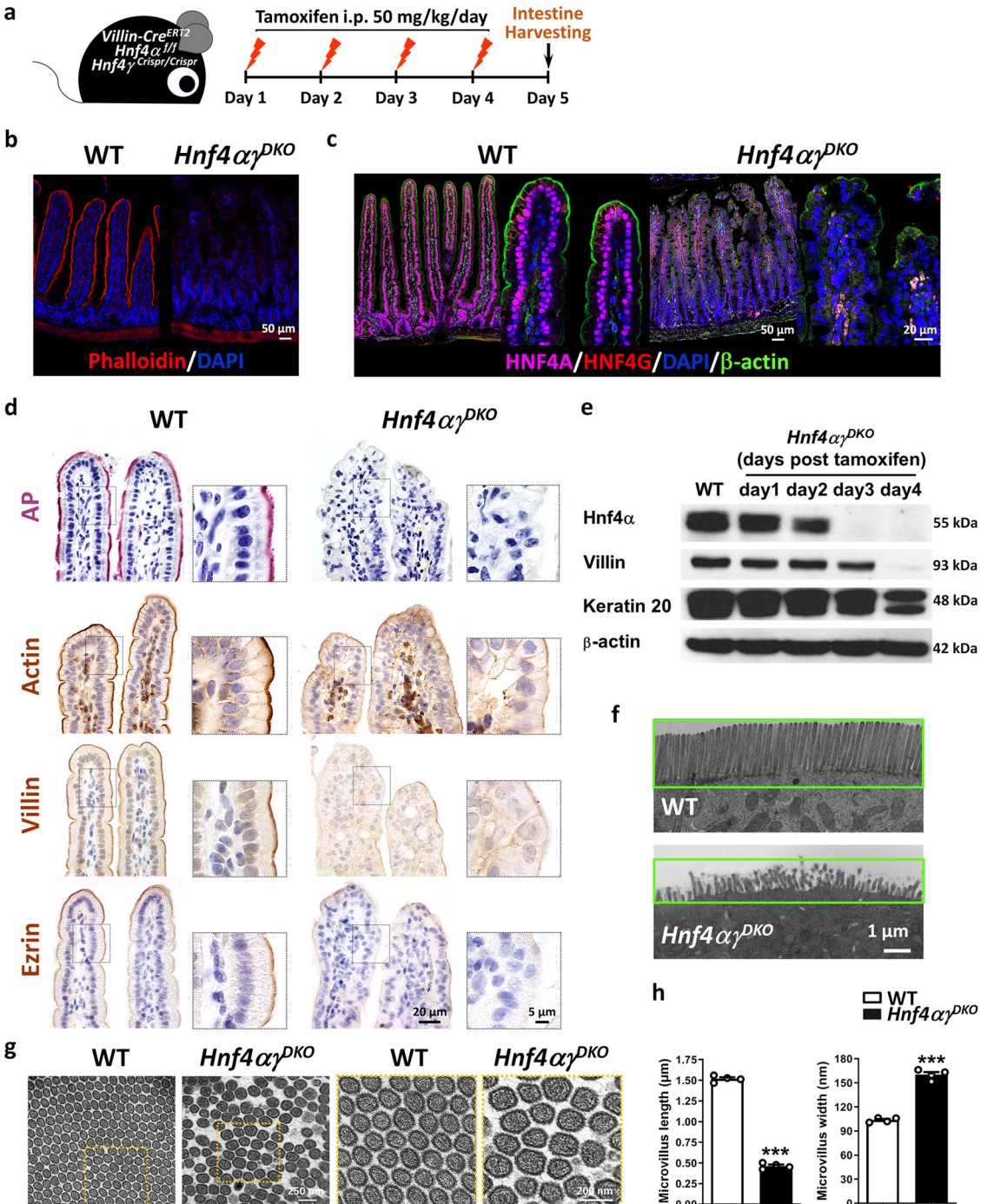

**Fig. 3 HNF4 transcription factors are required for intestinal brush border. a** Schematic of experimental design. **b** Phalloidin staining ($n = 3$ biologically independent mice). **c** Co-staining of HNF4A, HNF4G, and β-actin (brush border marker) by immunofluorescence confocal microscopy ($n = 3$ biologically independent mice). Individual images are shown in Supplementary Fig. 5a. **d** Brush border markers, such as alkaline phosphatase (AP, pink color), β-actin (brown color), Villin (brown color), and Ezrin (brown color) are localized to the apical surface of villus enterocytes and normally expressed in WT. Compromised expression levels of these brush border markers were observed in $Hnf4\alpha\gamma^{DKO}$ mice after 4 days of tamoxifen injection ($n = 3$ biologically independent mice). **e** Brush border markers (e.g., Villin and Keratin 20) were decreased in a time-dependent manner upon HNF4 loss, as evidenced by western blotting ($n = 2$ independent experiments). Brush border microvilli were notably shorter and disorganized (**f**) but wider (**g**) in $Hnf4\alpha\gamma^{DKO}$ mutants compared to WT mice, as evidenced by electron microscopy ($n = 4$ biologically independent mice, 4 days after tamoxifen injection). The high-magnification images are shown in the boxed insets. **h** Quantification of microvillus length and width. The data are presented as mean ± SEM ($n = 4$ biologically independent mice, Student's $t$-test, two-sided at ***$P < 0.001$).

**HNF4A is required for embryonic yolk sac brush border.** In addition to intestinal epithelium, the brush border can also be found in murine embryonic yolk sac epithelium[30]. Unlike the intestinal epithelium, *Hnf4g* is not detectable in the yolk sac

(Fig. 5a). HNF4A is expressed throughout the yolk sac epithelium, both in the villous tissue proximal to the placental disc and in smoother epithelial tissue in more distal areas (Fig. 5b). To study the function of HNF4 in the embryonic yolk sac, we crossed

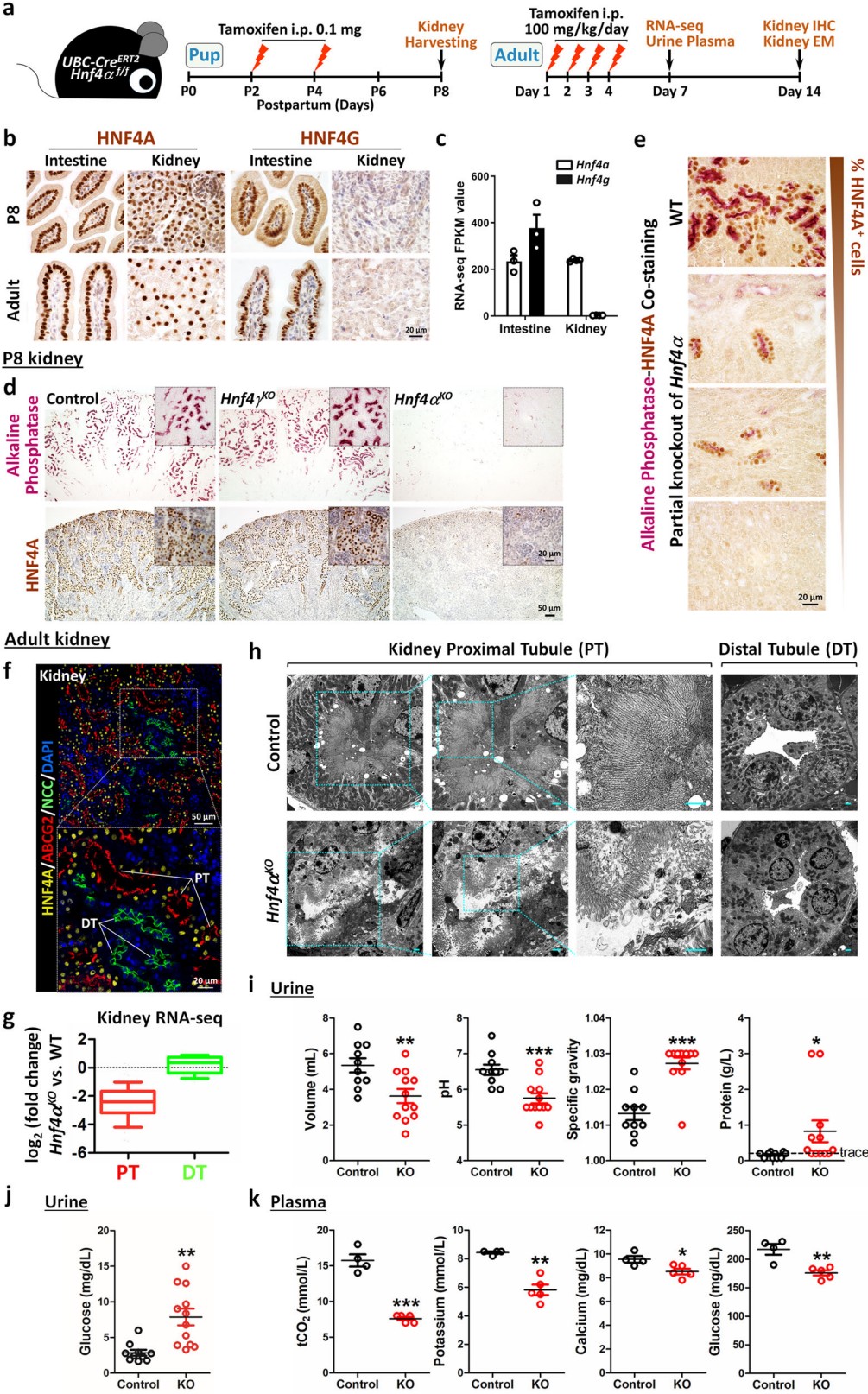

female *Hnf4α^{f/f}* with male *UBC-Cre^{ERT2}; Hnf4α^{f/f}* and knocked out HNF4A in the embryonic yolk sac by gavaging tamoxifen in pregnant female mice at embryonic day (E) 14.5 (Fig. 5c). Yolk sac tissues were collected at E17.5 and exhibited a clear disruption of brush border markers in *Hnf4α^{KO}* tissues (Fig. 5d). The yolk sac brush border plays an important role in nutrient transport

during embryonic development. To achieve a higher fraction of HNFA4-negative yolk sac tissues and assay for yolk sac functions, we treated pregnant mice with a mixture of tamoxifen and progesterone at E12.5 and E13.5 (Fig. 5e). We found that *Hnf4α^{KO}* embryos showed reduced size (Fig. 5f) and weight (Fig. 5g) compared to their *Cre*-negative littermate controls, suggesting a

**Fig. 4 HNF4A is required for renal brush border. a** Schematic of experimental design. **b** IHC staining of HNF4A and HNF4G in mouse intestine and kidney at 8 days postnatal (P8) or adult stages ($n = 3$ biologically independent mice). **c** Transcript levels of *Hnf4a* and *Hnf4g* in adult intestine and kidney (RNA-seq, $n = 3$ intestine and 4 kidney biological samples). **d** Alkaline phosphatase and IHC of HNF4A staining in P8 kidney. Compared to controls, alkaline phosphatase activity is only compromised in *Hnf4α*[KO] but not in *Hnf4γ*[KO] mutants ($n = 3$ biologically independent mice). **e** Co-staining of alkaline phosphatase and HNF4A in P8 kidney. Alkaline phosphatase activity is restricted to cells expressing HNF4A ($n = 3$ biologically independent mice). **f** Co-staining of HNF4A, ABCG2 (ATP-binding cassette transporter, a proximal tubule marker), and NCC (sodium chloride cotransporter, a distal tubule marker) by immunofluorescence confocal microscopy in adult kidney ($n = 3$ biologically independent mice). Individual images are shown in Supplementary Fig. 6a. PT: proximal tubule; DT: distal tubule. **g** Boxplots of kidney RNA-seq data show that proximal tubule transcripts are reduced, whereas distal tubule transcripts are not compromised in *Hnf4α*[KO]. The box represents the first and third quartiles with the horizontal line showing the median; whiskers represent the 10th and 90th percentile (Mann–Whitney test, PT vs. DT of $\log_2$ fold-change (*Hnf4α*[KO] vs. WT), two-sided at $P < 0.001$, $n = 4$ biologically independent mice, 7 days after tamoxifen injection). **h** Disorganized brush border was observed in proximal tubules of adult kidney upon HNF4A loss, whereas the distal tubules are less affected, as observed by electron microscopy (scale bar: 1 μm; $n = 2$ biologically independent mice, 14 days after tamoxifen injection). The high-magnification images are shown in the boxed insets. Symptoms of kidney dysfunction were observed upon HNF4A loss, as evidenced by (**i, j**) urine and (**k**) plasma assays. For urine samples, $n = 10$ controls and 12 mutants; for plasma samples, $n = 4$ controls and 5 mutants. The data are presented as mean ± SEM (7 days after tamoxifen injection, Student's *t*-test, two-sided at ***$P < 0.001$, **$P < 0.01$, and *$P < 0.05$).

---

disrupted brush border upon HNF4A loss (Fig. 5d) may contribute to these underdeveloped embryos. GSEA analysis confirmed that brush border genes were significantly downregulated in the yolk sac upon HNF4A loss (Fig. 5h and Supplementary Fig. 10a), which is consistent with the findings in the intestine and kidney upon HNF4 loss (Fig. 2d and Supplementary Fig. 3). The amniotic fluid is encased by the yolk sac and provides a supportive environment for the developing embryos. As brush border markers and gene expression are disrupted upon HNF4A loss in the yolk sac, we wondered whether there were corresponding deficiencies in the adjacent amniotic fluid. During pregnancy, amino acids represent one of the major nutrients for embryos and we found both essential and non-essential amino acids were all reduced in the amniotic fluid upon HNF4A loss, as evidenced by liquid chromatography–mass spectrometry (LC-MS) analysis (Fig. 5i and Supplementary Fig. 10b). Coincidentally, amino acid transport-related transcripts were also downregulated in *Hnf4α*[KO] yolk sac (Fig. 5j, k and Supplementary Fig. 10c), implying defective nutrient transport upon HNF4A loss. Furthermore, compared to *Cre*-negative littermate controls, we found increased uric acid in the amniotic fluid (Fig. 5l) and compromised expression of uric acid transporters in the yolk sac of *Hnf4α*[KO] (Fig. 5m, n and Supplementary Fig. 11), suggesting waste excretion might also be impaired in the yolk sac of *Hnf4α*[KO]. Together, these findings suggest that HNF4 is required for proper brush border assembly and functions of the yolk sac.

**HNF4 loss leads to increased stress fiber formation.** All together, we found disrupted brush borders of the intestine, kidney, and yolk sac upon HNF4 loss. Actin filaments provide microvilli structural support of brush border under normal conditions. Stress fibers are higher-order cytoskeletal structures consisting of actin filaments, crosslinking proteins, and myosin motors, and they counter membrane tension and stabilize cell structure by generating force and transducing mechanical signaling[31]. In most cases, stress fibers connect to focal adhesions and play an important role in mechanotransduction[32]. We wondered whether disrupted brush border could promote redirection of actin monomers away from the membrane and into stress fibers instead. In the yolk sac, we found upregulation of stress fiber and focal adhesion gene signatures upon HNF4A loss (Fig. 6a and Supplementary Fig. 12a–c). For example, the keratin network (Keratin 8/18) plays a crucial role in stress fiber reinforcement[33] and mechanotransduction[34], whereas Laminin γ-2 functions in focal adhesion stability[35]. In addition, calpains also play an important role in focal adhesion and stress fiber formation[36]. The elevated protein levels of Keratin 18, Laminin γ-2, and Calpain 2 were all confirmed in the yolk sac epithelium upon HNF4A loss

(Fig. 6b and Supplementary Fig. 12d). Similarly, in the intestinal epithelium, we found upregulated expression of many stress fiber and focal adhesion markers (Fig. 6c, e), as well as increased chromatin looping events and active chromatin marks at these gene loci upon HNF4 loss (Fig. 6d). The filamin family is named for its filamentous colocalization with actin stress fibers and increased filamin was also observed in the intestinal epithelium of *Hnf4αγ*[DKO] (Fig. 6f–h), suggesting that alterations in architecture of brush border in *Hnf4αγ*[DKO] might result in a cytoskeleton reorganization with increased stress fiber formation (Fig. 6i).

Filamin A has two flexible hinge regions that are susceptible to proteolysis by calpains[37]. We found elevated transcript levels of Filamin A but not protein levels in the yolk sac tissues of *Hnf4α*[KO] (Supplementary Fig. 12e, f), suggesting posttranslational regulation or degradation of Filamin A proteins may take place. In the kidney, it seems the stress fiber markers, such as Calpain 2 and Filamin A, are more enriched in the distal tubules of the kidney (Supplementary Fig. 12g, h). Unlike the intestine and yolk sac, stress fibers are not detected in the brush border-containing cells of the kidney proximal tubules. In healthy murine kidneys, Keratin 18 is not expressed in proximal or distal tubules and is only localized in the collecting ducts[38]. Interestingly, Keratin 18 is observed de novo in distal tubules in kidney injury models[38] and we found upregulated Keratin 18 in the kidney of *Hnf4α*[KO] (Supplementary Fig. 12g), consistent with impaired kidney function upon HNF4A loss.

## Discussion

HNF4A has been reported to regulate intestinal epithelial barrier function, with destabilized intercellular junctions in the absence of HNF4A[39]. Chiba et al.[40] previously reported that overexpression of HNF4A could induce the formation of brush border in the F9 cells in vitro. In this study, we show that HNF4 functions as a conserved and important regulator of brush border genes in multiple tissues in vivo, including the intestine, kidney, and yolk sac, and demonstrate comprehensive transcriptional regulatory mechanisms of HNF4 on brush border gene expression. We found that HNF4 factors activate brush border gene expression through binding to distal enhancer regions, maintaining enhancer chromatin activity and facilitating chromatin looping in the intestine. The brush border dysfunction is an important aspect of enterocyte abnormalities in *Hnf4αγ*[DKO], although this *Villin-Cre*[ERT2] model could not exclude the effects of HNF4-deficient intestinal stem cells in this study.

Mechanosignaling transduces signals much faster than chemical stimulation[41]. When cells are under mechanical stress, actin stress fibers increase to reinforce their mechanical strength. Elevated membrane tension lowers actin based-protrusion[42]. Impaired brush border transcripts and enhanced stress fiber/focal

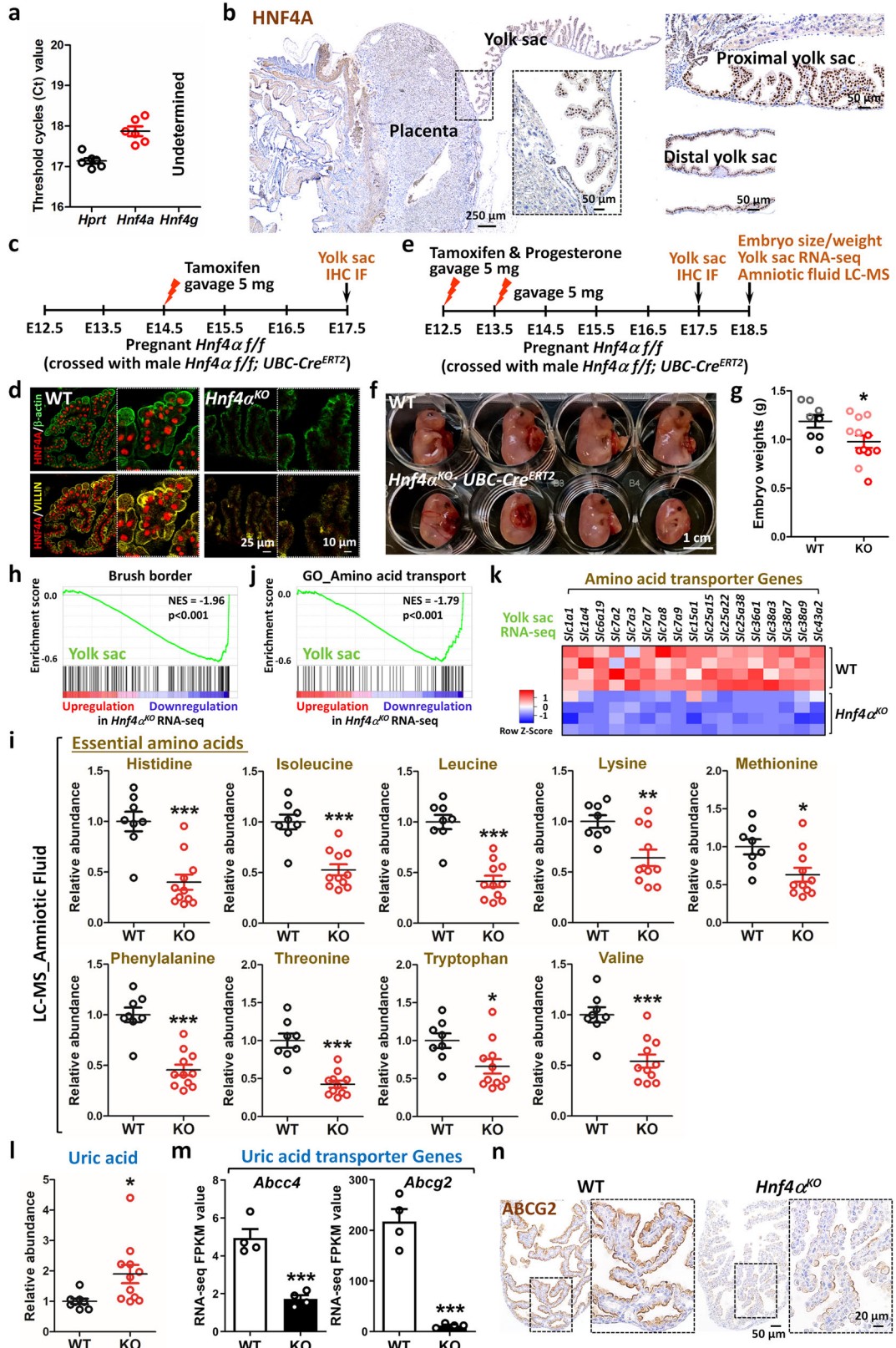

adhesion transcripts were observed upon HNF4 loss, suggesting that HNF4 transcription factors may work as a mechanosignaling sensor, and future study is needed to test this idea. It is also of interest to note that there is a substantial increase of Filamin A upon HNF4 loss in the intestinal epithelium. Filamin A undergoes proteolysis and its proteolysed fragments can localize to the nucleus. It has been reported that Filamin A inhibits tumor growth and metastasis by interacting with transcription factors when the active cleaved form of Filamin A localizes to the nucleus[43–45]. Rapid immunoprecipitation MS of endogenous proteins data show HNF4A can interact with Filamin A[46]; it will be interesting to investigate whether active cleaved form of

**Fig. 5 HNF4A is required for embryonic yolk sac. a** qPCR shows that *Hnf4a* is expressed in the yolk sac epithelium, whereas *Hnf4g* is not detectable (*n* = 6). **b** IHC staining of HNF4A (*n* = 6) in the yolk sac epithelium. **c, d** Immunofluorescence co-staining of HNF4A and brush border markers Villin and β-actin in the E17.5 yolk sac epithelium (*n* = 3 biological replicates), see schematic of experimental design in **c**. **e–n** Functional impairment analysis of yolk sac upon *Hnf4a* loss. **e** Schematic of experimental design. **f** *Hnf4α^KO* embryos show smaller size than littermate *Cre*-negative controls (*n* = 2 independent experiments). **g** Embryo weights (*n* = 8 controls and 12 mutants, Student's *t*-test, two-sided at *\*P* < 0.05). Circles represent biological replicates from two batches of experiments. Samples with red and black color are from one batch, whereas samples with light red and gray are from the other batch. **h** GSEA of RNA-seq data from E18.5 yolk sac reveals that the brush border gene transcripts are compromised upon HNF4A loss (Kolmogorov–Smirnov test, one-sided for positive and negative enrichment scores, *P* < 0.001). **i** Essential amino acids in amniotic fluid are reduced upon HNF4A loss, as evidenced by LC-MS. **j** GSEA of RNA-seq data from E18.5 yolk sac reveals that the amino acid transport gene signatures are compromised upon HNF4A loss (Kolmogorov–Smirnov test, one-sided for positive and negative enrichment scores, *P* < 0.001). **k** Heatmap shows examples of amino acid transporter genes[82] with significant downregulation upon HNF4A loss (*Hnf4α^KO* vs. WT at FDR < 0.05). Statistical tests were embedded in Cuffdiff, see bar charts in Supplementary Fig. 10c. LC-MS shows increased uric acid in amniotic fluid (**l**) upon HNF4A loss and, correspondingly, the uric acid transporter transcripts (**m**) are also compromised in the yolk sac of *Hnf4α^KO* embryos. For all LC-MS data (**i, l**), one mutant sample of amniotic fluid was excluded due to the overall undetectable signals for most of the metabolites (*n* = 8 controls and 11 mutants). Normalized ion counts were further normalized relative to the average abundance of metabolite from WT littermates of the same experimental batch. The data are presented as mean ± SEM (Student's *t*-test, two-sided at \*\*\**P* < 0.001, \*\**P* < 0.01, and *\*P* < 0.05). For RNA-seq data of E18.5 yolk sac, *n* = 4 controls and 4 mutants (\*\*\*Cuffdiff FDR < 0.001). **n** IHC staining of ABCG2 (brush border marker; uric acid transporter) in the E17.5 yolk sac epithelium (*n* = 4 biological replicates).

Filamin A could activate HNF4 transcription factors to form a positive feedback of brush border gene expression in the future.

Deletion of HNF4A (embryonic-onset) in nephron progenitor cells by *Six2GFPcre* (mosaic expression) leads to a defect in the development of proximal tubules, as evidenced in the kidney of E18.5[26]. Less clear is whether adult-onset disruption of HNF4A would compromise kidney function. Importantly, our study demonstrates compromised proximal tubules and impaired reabsorption in an adult-onset knockout of HNF4A, recapitulating Fanconi syndrome. Approximately 93% of genes that are significantly altered in the developing kidney upon HNF4A loss (*Six2GFPcre*, FDR < 0.05)[26] also show significant changes in the adult kidney upon HNF4A loss (*UBC-Cre^ERT2*, FDR < 0.05). Along with these findings, an R76W mutation in human *HNF4A* is also linked to Fanconi syndrome[47]. Defective proximal tubule function causes Fanconi syndrome and the adult-onset knockout of *Hnf4a* could be used to study Fanconi syndrome. In addition, Wnt signaling patterns the proximal-distal axis of the nephron and bone morphogenetic protein (BMP) antagonizes Wnt gradient[48,49]. The formation of the proximal tubules requires Notch signaling and an environment of low Wnt signaling[49–53]. Recently, a reinforcing feed-forward module between HNF4 and BMP signaling was shown to promote enterocyte differentiation[15]. Future investigation is needed to explore whether the interactions of BMP/Wnt/Notch signaling and HNF4 transcriptional regulatory networks are conserved in the development of the proximal tubules of kidney.

## Methods

**Mice and treatment**. Laboratory mice (*Mus musculus*) were housed in a room with controlled temperature of 21–23 °C and humidity of 30–70% under a 12 h light/12 h dark cycle. The *Villin-Cre^ERT2* transgene[54], *UBC-Cre^ERT2* transgene[55], *Hnf4α^f/f* [56], and *Hnf4γ^Crispr/Crispr* [15] alleles were integrated to generate the conditional compound mutants and controls. All mouse protocols and experiments conducted had the approval of the Rutgers Institutional Animal Care and Use Committee. To avoid circadian variability, we collected samples between noon to 2 pm.

To study intestinal brush border, experimental *Villin-Cre^ERT2* mice (8–12 weeks old) were administered with tamoxifen (Sigma T5648) at 50 mg/kg/day or vehicle, intraperitoneally (i.p.). Histologic and western blot samples were collected from mice after 4 consecutive days of tamoxifen (to induce *Cre* recombination) or vehicle treatment. H3K4me3 HiChIP-seq and H3K27ac MNase ChIP-seq samples were collected from mice after 3 consecutive days of tamoxifen or vehicle treatment. RNA-seq samples were collected from mice after 2 or 3 consecutive days of tamoxifen or vehicle treatment.

To study renal brush border in pups, experimental *UBC-Cre^ERT2*-positive and -negative pups were all treated with tamoxifen at 0.1 mg/day by i.p. injection 2 days (P2) and 4 days (P4) after their birth, and kidney tissues were collected at day 8 after their birth (P8) for further analysis. See schematic of experimental design in Fig. 4a, left panel.

To study renal brush border in adult, experimental *UBC-Cre^ERT2*-positive and -negative mice (5 weeks old) were all treated with tamoxifen at 100 mg/kg/day by i. p. injection for 4 consecutive days. Renal histologic analysis was done 7 days or 14 days after tamoxifen treatment. Kidney (RNA-seq), urine, and plasma were collected 7 days after tamoxifen treatment. See schematic of experimental design in Fig. 4a, right panel.

To study brush border of yolk sac, pregnant *UBC-Cre^ERT2*-negative female (crossed with *UBC-Cre^ERT2*-positive male) were treated with (1) 5 mg tamoxifen by oral gavage at E14.5 or (2) treated with a mixture of tamoxifen and progesterone (Sigma P0130) at 5 mg by oral gavage at E12.5 and E13.5 or 3) treated with a mixture of tamoxifen and progesterone at 5 mg by oral gavage at E12.5, and followed by 2 mg via i.p. injection at E13.5. Here, progesterone was co-administered with tamoxifen (1 : 1 mix, 5 mg each) to prevent fetal abortions in pregnant mice[57]. See schematic of experimental design in Fig. 5c, e and Supplementary Fig. 11b.

**Intestinal epithelium isolation**. Intestine was collected, flushed with cold phosphate-buffered saline (PBS), and opened longitudinally. After cutting the intestine into 1 cm pieces, the tissues were rotated in 3 mM EDTA/PBS at 4 °C for 5 and 10 min. The EDTA/PBS was refreshed every time. After a light shake, the WT and mutant tissues were transferred to tubes with fresh 3 mM EDTA/PBS, respectively. The time of EDTA incubation was adjusted to release all the epithelial cells from underlying muscular layer. WT tissues were incubated for 40 min, whereas mutant tissues were incubated for 20 min. The intestine tissues were vigorously shaken and the whole epithelium fraction was collected from the supernatant. Villus and crypt cells were separated using a 70 μm cell strainer, where crypt cells passed through and villus cells were kept on the top of the strainer. Cells were pelleted by centrifugation at 4 °C, 200 × *g* for 3 min. After washing with cold PBS, cell pellets were used for experiments as described in later sections.

**Histology and staining**. Intestinal, renal, and yolk sac tissues were fixed in 4% paraformaldehyde at 4 °C overnight, washed in PBS prior to dehydration and paraffin embedding. For cryo-embedding, after fixation and PBS wash, tissues were processed with 15% sucrose for 4 h and 30% sucrose until tissues sunk. Optimal cutting temperature (OCT) compound (Tissue-Tek 4583) was used for cryo-embedding. Five-micrometer paraffin sections were prepared for immunohistochemistry, immunofluorescence, and AP staining. Ten-micrometer cryosections of duodenal epithelium were prepared for phalloidin staining (Thermo Fisher Scientific A34055) according to the manufacturer's instructions. AP is an enzyme colocalized with brush border and its activity was detected using the AP Staining Kit II (Stemgent). Periodic Acid–Schiff staining was used to detect proximal tubules in the kidney and slides were incubated in 0.5% periodic acid and stained with Schiff's Reagent (J612171, Alfa Aesar). Co-staining of AP and HNF4A was performed by AP staining and followed by HNF4A immunohistochemistry. Immunohistochemistry was performed using primary antibodies against Hnf4α (Santa Cruz sc-6556 X, 1 : 2000), Hnf4γ (Santa Cruz sc-6558 X, 1 : 2000), β-actin (Abcam ab8227, 1 : 1000), Villin (Santa Cruz sc-7672, 1 : 500), Ezrin (Cell Signaling #3145, 1 : 1000), Abcg2 (also called Bcrp, Kamiya Biochemical Company MC-980, 1 : 500), Cytokeratin 18 (Santa Cruz sc-51582, 1 : 200), Calpain 2 (Santa Cruz sc-373966, 1 : 200), Laminin γ-2 (Santa Cruz sc-28330, 1 : 200), and Filamin A (Abcam ab76289, 1 : 500). Blocking reagents and secondary antibodies were used according to the manufacturer's protocol of the Vectastain ABC HRP Kit (Vector Labs). DAB (0.05%; Amresco, #0430) and 0.015% hydrogen peroxide in 0.1 M Tris were used for immunohistochemistry developing. The slides were counterstained with hematoxylin, mounted, and viewed on a Nikon Eclipse E800 microscope. The

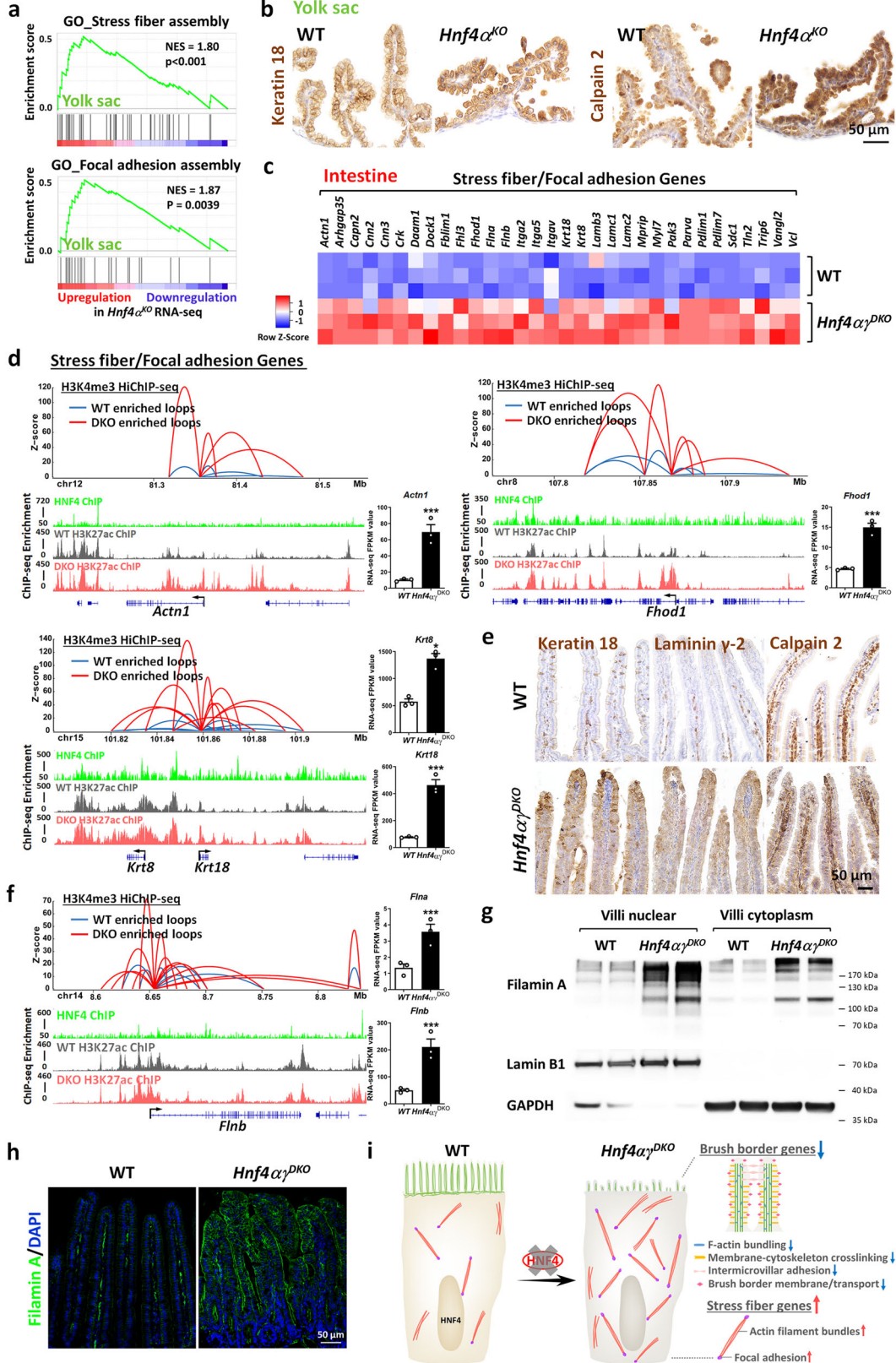

staining was imaged with a Lumenera INFINITY3 camera and infinity capture imaging software (v6.5.6). A Zeiss Axiovert 200 M fluorescence microscope was used for imaging phalloidin staining using a Retiga-SRV CCD (QImaging) and QCapture imaging software (v2.0.8). A Zeiss LSM 510 Meta confocal microscope and Carl Zeiss ZEN 2009 (v5.5.0.451) software were used for imaging the immunofluorescence staining of Hnf4α (Abcam ab41898, 1 : 100), Hnf4γ (Santa Cruz sc-6558 X, 1 : 100), β-actin (Abcam ab8227, 1 : 100), Abcg2 (Kamiya Biochemical Company MC-980, 1 : 100), NCC (also known as Thiazide-Sensitive NaCl Cotransporter, Sigma AB3553, 1 : 100), Filamin A (Abcam ab76289, 1 : 100), and DAPI (Biotium 40043, 1 : 5000). ImageJ software (v1.52) was used for staining quantification.

**Transmission electron microscopy.** Freshly dissected intestinal and renal tissues were cut into 1 mm fragments and immediately fixed in 0.1 M sodium cacodylate buffer (pH 7.4, Electron Microscopy Sciences 11653) containing 2%

**Fig. 6 HNF4 loss results in an increased stress fiber formation. a, b** Yolk sac data of WT vs *Hnf4α$^{KO}$*. Schematic of experimental design is shown in Fig. 5e. **a** GSEA of RNA-seq data from E18.5 yolk sac reveals that stress fiber assembly (Kolmogorov–Smirnov test, one-sided for positive and negative enrichment scores, $P < 0.001$) and focal adhesion assembly (Kolmogorov–Smirnov test, one-sided for positive and negative enrichment scores, $P = 0.0039$) gene signatures are all elevated upon HNF4A loss. **b** IHC staining of stress fiber/focal adhesion-related proteins ($n = 3$ biological replicates) in E17.5 yolk sac. **c–i** Intestinal epithelium data of WT vs. *Hnf4αγ$^{DKO}$*. **c** Heatmap of RNA-seq data shows upregulated stress fiber/focal adhesion transcripts upon HNF4 loss in the intestinal epithelium (FDR < 0.05, $n = 3$ biologically independent mice). **d** Increased chromatin loops were observed at stress fiber/focal adhesion gene loci upon HNF4 loss. H3K4me3 HiChIP-seq was done in villus cells of *Hnf4αγ$^{DKO}$* and their littermate controls. Differential loops (DEseq2 $P < 0.05$) are visualized by Sushi package for the loops with $q \leq 0.0001$ and counts $\geq 8$ (combined 2 replicates). Bar charts show transcript levels of brush border genes. The data are presented as mean ± SEM (RNA-seq: $n = 3$ biological replicates, ***Cuffdiff FDR < 0.001 and *FDR < 0.05). H3K4me3 HiChIP-seq: $n = 2$ biological replicates; H3K27ac ChIP-seq: $n = 2$ biological replicates; HNF4 ChIP-seq: $n = 2$ biological replicates. **e** IHC staining of stress fiber/focal adhesion-related proteins ($n = 3$ biological replicates). **f** Increased chromatin loops at *Flnb* locus were observed upon HNF4 loss. Elevated protein levels of Filamin A were observed by **g** western blotting ($n = 2$ biologically independent mice) and **h** immunofluorescence staining ($n = 3$ biologically independent mice). **i** Loss of HNF4 factors results in downregulated brush border genes and upregulated stress fiber genes.

paraformaldehyde (Electron Microscopy Sciences 15714-S) and 2.5% glutaraldehyde (Electron Microscopy Sciences 16216) at 4 °C overnight. Tissue processing, embedding, sectioning, and imaging were performed using standard procedures as described[58].

**Protein extraction and western blotting**. RIPA buffer (50 mM Tris-HCl pH 8.0, 0.1% SDS, 150 mM NaCl, 0.5% Na-deoxycholate, 1% NP-41, phosphatase inhibitors, and protease inhibitor cocktails) was used to extract the total protein. The protein from nuclear and cytoplasmic fractions was separated as described[59]. Bioruptor sonication was used to break cells from duodenal epithelium before (4 cycles, 30 s on, and 30 s off) and after (2 cycles, 30 s on, and 30 s off) rotating in lysis buffer at 4 °C for 30 min, respectively. Pierce BCA Protein Assay Kit (Thermo) was used to measure protein concentration. The following primary antibodies were used in this study: Hnf4α (Santa Cruz sc-6556 X, 1 : 1000), Villin (Santa Cruz sc-7672, 1 : 1000), Keratin 20 (Cell Signaling #13063, 1 : 2500), Filamin A (Abcam ab76289, 1 : 1000), Lamin B1 (Abcam ab16048, 1 : 1000), GAPDH (Santa Cruz sc-25778, 1 : 5000), and β-actin (Abcam ab8227, 1 : 5000). The blots with protein ladders can be found in the Source Data file.

**RNA extraction and quantitative reverse transcription-PCR**. Yolk sac tissues were processed for RNA extraction using Trizol (Invitrogen) according to the manufacturer's instructions. cDNA was synthesized from total RNA with Oligo (dT)$_{20}$ primers using SuperScript III First-Strand Synthesis SuperMix (Invitrogen). Quantitative reverse transcription-PCR was performed to measure changes in mRNA expression using Applied Biosystems 7900HT Sequence Detection System with Power SYBR® Green PCR Master Mix. The amplification conditions were as follows: 2 min at 50 °C, 10 min at 95 °C, followed by 40 cycles of 15 s at 95 °C and 1 min at 60 °C. The primer sequences are provided in Supplementary Table 1. Hypoxanthine-guanine phosphoribosyl transferase 1 (*Hprt1*) was used as a housekeeping gene.

**Urine and plasma sample collection and assays**. Adult *UBC-Cre$^{ERT2}$* experimental mice (7 days post tamoxifen injection) were placed in individual metabolic cage overnight, with free access to food and water. Urine was collected in the next morning and was tested using Urine Reagent Strip (Teco Diagnostics URS-10) according to the manufacturer's instructions. To measure urine glucose, urine was filtered with a phospholipid removal column (Phenomenex 8B-S133-TAK) for deproteinizing and then measured using Glucose Assay Kit (abcam ab65333) according to the manufacturer's instructions. Blood was collected from abdominal aorta of adult *UBC-Cre$^{ERT2}$* experimental mice (7 days post tamoxifen injection) with lithium heparin (Sigma H0878). Blood samples were centrifuged at 1500 × *g* for 15 min at 4 °C and the supernatant was collected as the heparinized plasma. One hundred microliters of heparinized plasma was transferred to the VetScan Preventive Care Profile Plus reagent rotor (Abaxis 500-7185) and measured using Abaxis VetScan VS2 Chemistry Analyzer.

**Metabolite extraction of amniotic fluid and LC-MS analysis**. Amniotic fluid was collected from E18.5 yolk sac and stored at −80 °C for subsequent metabolite extraction, as described below. Amniotic fluid samples were centrifuged at 1500 × *g* for 10 min at 4 °C and the supernatant was used for metabolite extraction. Forty microliters of ice-cold methanol was added to 10 μl of amniotic fluid and samples were vortexed for 10 s, incubated for 20 min at −20 °C, and then centrifuged at 16,800 × *g* for 10 min at 4 °C. The supernatant was collected as the first extract. Two hundred microliters of 40 : 40 : 20 methanol : acetonitrile : H$_2$O was added to the remaining pellet and samples were vortexed for 10 s, incubated for 10 min on ice, and then centrifuged at 16,800 × *g* for 10 min at 4 °C. The supernatant was combined with previous extract. The mixed extract was transferred into a phospholipid removal column (Phenomenex 8B-S133-TAK) for further cleanup and then stored at −80 °C for subsequent LC-MS analysis. LC-MS analysis of the extracted metabolites was conducted as previously described[17].

**RNA-seq and data analysis**. Duodenal epithelial cells, kidney, and E18.5 yolk sac were processed for RNA extraction using Trizol (Invitrogen) according to the manufacturer's instructions. Duodenal epithelial cells were sequenced using Illumina's TruSeq RNA Library Prep kit v2 at Sequencing Facility of the Rutgers Cancer Institute of New Jersey, whereas kidney and yolk sac tissues were sequenced using MGISEQ-2000 at BGI-Shenzhen, China. To compare expression levels of brush border genes in different tissues, Kallisto[60] (v0.44.0) was utilized to quantify the transcript abundances of the RNA-Seq samples through *pseudoalignment*, using single-end or paired-end reads and a RefSeq mm9 transcriptome build index. Then, the tximport[61] (v1.8.0) package was run in R (v3.5.2) to create gene-level count matrices for use with DESeq2[62] (v1.20) by importing quantification data obtained from Kallisto. DESeq2 was then used to generate FPKM values per kilobase of gene length per million mapped fragments for the RNA-seq data of intestine tissues and kidney tissues. These FPKM values were further used to compare with the public RPKM values of embryonic yolk sac[20]. To compare transcript levels of mutants vs. controls, fastQC (v0.11.3) was used to check the quality of raw sequencing reads. Tophat2 (v2.1.0) was used to align the reads to the mouse (mm9) genomes and generate bam files. The cxb files were generate from these bam files using Cuffquant (v2.2.1) and FPKM values were calculated with Cuffnorm (v2.2.1) using quartile normalization. Differentially expressed genes were identified between the controls and the mutants using Cuffdiff (v2.2.1) with quartile normalization and per-condition dispersion. Genes with FPKM > 1 were used for RNA-seq-related analysis. RNA-seq data (bam files) were visualized using the Integrative Genomics Viewer[63] (IGV 2.4.13). The heatmaps were plotted using Heatmapper[64] with normalized FPKM values of RNA-seq. GSEA (v4.0.3)[65] was done on the preranked gene list. Gene Ontology analysis was done with DAVID[66] (v6.8). Brush border gene list used in this study was combined from following Gene Ontology terms, including brush border genes (GO:0005903), brush border membrane genes (GO:0031526), and microvillus genes (GO:0005902). The scRNA-seq data were re-analyzed and visualized by Seurat package[67,68] (v3.1.5).

**Data analysis of ChIP-seq and DNase-seq**. FastQC (v0.11.3) was used to check the quality of raw sequencing reads (fastq), and bowtie2 (v2.2.6) was used to align the sequences to mouse (mm9) genomes and generate bam files. MACS[69] (v1.4.1) was used for peak calling and to generate bed files from aligned reads. Hnf4 ChIP-seq (HNF4A, Santa Cruz sc-6556 X, lot B1015, 6 μg per ChIP; HNF4G, Santa Cruz sc-6558 X, lot F0310, 6 μg per ChIP) of mouse duodenal epithelium are at a *p*-value of 10$^{-3}$; DNase-seq of the intestine and kidney, Hnf4 ChIP-seq of kidney, and H3K27ac (Abcam ab4729, lot GR184332-2, 6 μg per ChIP) MNase ChIP-seq of mouse duodenal epithelium are at a *p*-value of 10$^{-5}$. The distribution of HNF4-binding sites was analyzed with the *cis*-regulatory element annotation system (CEAS, v1.0.0)[70]. BEDTools[71] (v2.17.0) was used to merge, intersect, or subtract the intervals of bed files. Peak2gene/BETA-minus[72] (v1.0.0) or GREAT analysis[73] (v3.0.0) was used to identify genes within from genomic regions (bed file). Deeptools bamCoverage[74] (v2.4.2, duplicate reads ignored, RPKM normalized, and extended reads) was used to generate bigwig files from bam files. BigWigMerge (UCSC, v2) was used to merge the bigwig files of different replicates. IGV was used to visualize normalized bigwig tracks. *k*-means clustering heatmaps of ChIP-seq/DNase-seq were created with Haystack[75] (v0.4.0) using quantile normalized bigwigs using computeMatrix and plotHeatmap from deeptools[74] (v2.4.2). Genomic regions of desired *k*-means clusters were extracted from bed files generated by plotHeatmap. DiffBind[25] (v2.4.7) was used to identify differential signals of Hnf4 ChIP-seq in the intestinal and renal tissues, and 0.01 was used as the cutoff for significance. SitePro[70] (v1.0.2) was used to plot the average signal profiles of ChIP-seq/DNase-seq. Homer findMotifsGenome.pl[76] (v4.8.3) was used to find transcription factor motifs. Promoter was defined as region from 2 kb upstream of the transcription start site (TSS) to 2 kb downstream of the TSS, whereas enhancer was defined by excluding the promoter region.

**HiChIP-seq and data analysis**. Duodenal villus cells were isolated as describe in the above section. Cells were cross-linked in 1.5% formaldehyde (Sigma F8775) at 4 °C for 10 min and then at room temperature for another 50 min. Cells were pelleted and

washed with ice-cold PBS twice by centrifugation at $300 \times g$, 4 °C. The HiChIP protocol[77] was conducted and H3K4me3 (Millipore 05-745 R, lot 3158071, 9 μL per ChIP) antibody was used as described[78]. HiC-Pro pipeline[79] (v2.11.1) was used to process the HiChIP data as described[78]. Bowtie2 (v2.3.4.3) was used to align reads to the mouse (mm9) genome assembly. DESeq2[62] (v1.20) was applied to identify differential loops using sequencing counts of H3K4me3 HiChIP-seq data. To identify the differential loops, we took hichipper[77] (v0.7.0) loops with raw counts ≥4 in both replicates and $q \le 0.0001$ in at least one replicate, from at least one of the two conditions being compared. HiChIP loops were visualized using Sushi package[80] (v1.20.0). For simplicity, biological replicates were combined for Sushi looping visualization.

**Statistical analysis**. The data is presented as mean ± SEM (GraphPad Prism v8.4.3) and statistical comparisons were performed using Student's t-test at ***$P < 0.001$, **$P < 0.01$, or *$P < 0.05$. Source data are provided as a Source Data file. The exact P-values are also shown in the Source Data file. Kolmogorov–Smirnov test was used on GSEA. Mann–Whitney test was used as part of RNA-seq analysis. Bioinformatics-related statistical analysis was done with the embedded statistics in each package, including Cuffdiff, DiffBind, HOMER, GSEA, DESeq2, DAVID, and hichipper. $P < 0.05$ (95% confidence interval) was considered statistically significant.

**Reporting summary**. Further information on research design is available in the Nature Research Reporting Summary linked to this article.

## Data availability

All the kidney and yolk sac RNA-seq data have been deposited in the Gene Expression Omnibus (GEO) database under accession codes: GSE157911 and GSE166125. All the HiChIP-seq data of this study have been deposited in the GEO database under accession code: GSE148691[78]. The villi ATAC-seq, Hnf4 mutants vs. WT transcriptome, HNF4 ChIP-seq, and H3K27ac MNase ChIP-seq of mouse intestinal epithelial cells have been deposited in the GEO database under accession code: GSE112946[15]. GSE47192[81] was used to reanalyze renal HNF4A ChIP-seq. GSE57919[23] and GSE51336[24] were used to reanalyze DNase-seq of the duodenum and kidney. PRJEB18767[20] was used to reanalyze RNA-seq of yolk sac. GSE112828[26] was used to reanalyze the transcriptome of developing kidney upon HNF4A loss for the comparison with our adult kidney data. Source data are provided with this paper.

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

## Acknowledgements

This research was funded by grants from the NIH (R01CA190558 and R01DK121915 to M.P.V. and R01DK112782 to A.S.). M.P.V. is also supported by the Intestinal Stem Cell Consortium from the National Institute of Diabetes and Digestive and Kidney Diseases (NIDDK) and National Institute of Allergy and Infectious Diseases (NIAID) of the National Institutes of Health under grant number U01 DK103141. The content is solely the responsibility of the authors and does not necessarily represent the official views of the National Institutes of Health. L.C. was supported by New Jersey Commission on Cancer Research grant (DFHS18PPC051). Support was also received from the Sequencing Facility and Metabolomics Shared Resource of the Rutgers Cancer Institute of New Jersey (P30CA072720) and imaging core facility of Human Genetics Institute of New Jersey. S.L., A.D., R.P.V., A.P., R.A., R.M., and N.H.T. were supported by MacMillan Summer Undergraduate Research Fellowships. We thank Eileen White for sharing the *UBC-Cre^ERT2* mice. We thank Lauren Aleksunes for helping us understand yolk sac biology. We thank Noriko Goldsmith for helpful imaging support. We also thank Anbo Zhou for his extra efforts, especially with helping us overcome difficulties during the pandemic.

## Author contributions

L.C. conceived and designed the study, performed benchwork, sequencing data processing, and bioinformatics, collected and analyzed the data, and wrote the manuscript. S.L. contributed to benchwork and Diffbind analysis. A.D., R.P.V., A.P., R.M., J.H., N.H.T., E.C., M.Y., W.C., J.F., N.G, A.S., X.S., and E.M.B. contributed to benchwork. R.A. and C.E.E. contributed to sequencing data processing. M.P.V. conceived, designed, and supervised the study, and wrote the manuscript.

## Competing interests

The authors declare no competing interests.
