## [Peer Review File · Nature Communications]

Reviewers' Comments:

Reviewer #1:

Remarks to the Author:

How transcription factors are involved in programmes that generate sub-cellular structures is an interesting and important question. The brush border is an important cellular structure in tissues involved in fluid intake, such as the intestine or kidney. In this study, the authors provide compelling evidence that HNF4 is critical for the generation of the brush border in three different tissues, intestine, kidney and yolk sack. Thus, HNF4 seems to be the master regulator of brush border genesis. This work also reveals a potential model system for diseases such as Fanconi syndrome. The data are of high quality and the paper is very well presented. What is also really nice is the use of various complementary approaches to strengthen conclusions, such as ChIP-seq, HiCHIP, RNA-seq. Thus, this paper provides also interesting resources for the community.

My only major comment is as follows:

While the study shows compellingly that HNF4 is essential for correct brush border formation in various tissues, it leaves open the exciting question: Is HNF4 sufficient to 'induce' brush border generation?

I think it would not be fair to ask to address this question in the mouse model, but I can imagine that one can devise a simple trans-(over)expression test in a cell culture system, e.g. using some fibroblast cell line that normally does not have a clear brush border. If such experiments do not provide evidence that HNF4 is sufficient to induce the program, that is ok and can be simply reported and figures shown as supplementary figures. However, if (over) expression of HNF4 causes the generation of some kind of brush border in a cell line that normally does not have it, this would be fantastic and increase the impact of this work further.

The authors could squeeze a bit more out of their rich data by performing some more bioinformatic analysis:

From their ChIP-seq data, to what extent are HNF4 binding sites found at putative enhancer elements (more distal or downstream of genes) as opposed to promoter elements (please, note that H3K27ac is found at both enhancers and promoters)?

The motif analysis (Fig. 1g) highlights also CTCF/CTCFL which chimes with their finding of a role of HNF4 in having some role to gene loops. What is the relationship between HNF4 and CTCF binding sites at brush border genes?

Fig. S9 should possibly move to the main figures, as this illustrates an interesting downstream effect of losing the brush border organisation. As supplementary figure, it is a bit hidden.

Fig.3 C: Another effect of deletion of HNF4 seems to be loss of order of the intestinal epithelial cells. In WT, the cells are nicely aligned in rows, this is completely lost in the KO. Maybe this is something the authors should point out in their text, too.

Legend of Fig 1, (I) typo: 'intetsine' to 'intestine', please.

Reviewer #2:

Remarks to the Author:

Chen and colleagues here take a multi-omic approach to defining the biologic targets of HNF4 in brush border containing tissues intestine, yolk sac and kidney. Using RNA-seq and DNase-seq, they show that all three tissues commonly express most brush border genes, and that HNF4 binding motifs are the most accessible of any transcription factor in the brush border geneset. ChIP-seq of HNF4A in kidney and intestine were 90% similar, suggesting a common gene regulation program across organs. Inducible knockout mice were generated and RNA-seq suggested a largely common set of downregulated genes in the KO enriched for brush border and

transport genes and most of these are bound by HNF4A by ChIP-seq.

Using H3K27ac-MNase-ChIP-seq and H3K4me3 HiChIP, the authors provide evidence that KO of HNF closes chromatin and disrupts looping in brush border genes. They then characterize the morphologic and functional consequences of HNF4A KO across tissues.

This is an interesting and technically well executed study. The strength lies in the multi-omic application of transcriptome and epigenome profiling techniques and linking these back to function, ie a common brush border gene regulatory program.

There are several things that would improve the manuscript:

1. While the multi-omic data is overall convincing, there is no validation of HNF4A targets by ChIP-seq in the KO tissues. Inclusion of validation data, even if simply select genes, would strengthen the work.
2. Fanconi syndrome is defined by glycosuria and amino aciduria as well as phosphate wasting. At a minimum, urine glucose should be measured though urinary phosphate would also be helpful.
3. The authors do point out that HNF4A has been knocked out in developing kidney but not in adult. It could be interesting to compare the magnitude of gene expression changes in Marable et al. with the results here. Presumably there would be fewer changes with the author's own dataset which helps to distinguish the utility of this dataset over the developmental one.

Minor:

1. It would be nice to see an analysis of stress fiber formation in KO kidney as well.
2. Can the authors present a fuller analysis of the genes with altered expression in the KO? Fig. 2C seems to selectively analyze only the brush border gene set.

Reviewer #3:

Remarks to the Author:

Chen et al. report a novel role of HNF4 in regulating expression of genes involved in formation of epithelial brush border in the intestine, kidney, and yolk sacs. Epithelial-specific silencing of Hnf4 expression perturbed epithelial microvilli architecture in the intestine, kidney, and yolk sac while also promoting expression of genes involved in control of actin stress fibers and focal adhesions in the intestine. Mechanistically, it is proposed that HNF4 modulates the transcription of brush border genes by influencing the formation of chromatin loops when HNF4 is associated with enhancer chromatin structure, thereby resulting exposure of active promoter regions of these genes.

The overall role of HNF4 in the regulation of gene programs for microvilli and stress fibers formation in intestinal epithelial cells is novel and interesting. While most of the data supports the proposed mechanisms, the study would be strengthened by a few additional experiments.

Minor Concerns:

1. In Fig. 2G, transcription of all genes analyzed (*Espn*, *Ezr*, *Myo7b*, and *Slc6a19*) is decreased in *Hnf4agDKO* mice. Please include the expression of genes other than brush border genes as a negative control.
2. In Fig. 3B, beta-actin expression in the apical compartment of epithelial cells should be quantified. Beta-actin expression seems to be reduced in immunofluorescence images (Fig.3B), but not in the western blotting results (Fig. 3D). Please elaborate on this discrepancy. To show F-actin organization, please consider presenting images after phalloidin staining and provide western blotting and imaging results showing other brush border proteins such as ezrin, villin and alkaline phosphatase.
3. The authors demonstrated expression of stress fibers/focal adhesions genes in the absence of HNF4. However, only upregulation of filamin protein expression in the intestine is shown. Please analyze expression of filamin and other focal adhesion proteins in the kidney and yolk sac

epithelium of HNF4a KO mice.

4. Please move Fig.S9 to the main figure as it provides important results related to stress fibers/focal adhesions in intestinal epithelial cells of Hnf4ag DKO mice. Discuss how focal adhesion genes are upregulated in Hnf4a DKO mice. Since HNF4a has been reported to regulate intestinal epithelial barrier function, a discussion of the relationship between HNF4 and genes involved in controlling perijunctional actinomyosin ring and intercellular junctions will be useful.

5. In Fig.5H, yellow lines are difficult to see. Please change the color.

6. The word "intestine" is misspelled in line 535. Please modify.

Reviewer #4:

Remarks to the Author:

The brush border is a unique structure found in limited organs, e.g. intestine, kidney and yolk sac, and contributes to the absorption and transport of substances by dramatically increasing the surface area. This research group previously reported that transcription factor HNF4 is important for ISC maintenance (*Gastroenterology*. 2020 Mar; 158 (4): 985-999), and intestinal epithelial identity (*Nature genetics* 51, 777-785 (2019)). Here, the authors showed that HNF4 is also important for the maintenance of brush border in the intestine, kidney and yolk sac. HNF4 triggers the brush border gene program through binding to distal enhancer regions and facilitating chromatin looping at the gene loci of brush border genes in the intestine. This manuscript contains some interesting findings, but several major concerns are raised.

Comments:

Line 21: "many" is not appropriate. Use "multiple" instead of "many".

Lines 25-26: "By ChIP-seq and RNA-seq, we find that HNF4 transcription factors bind to and activate the brush border gene program." The authors should clearly indicate the organ they analyzed. If they haven't analyzed all three organs

Lines 26-30: "To refine our understanding of HNF4-dependent regulation of brush border genes at the 3D chromatin architecture level, we apply H3K4me3 HiChIP-seq technology..." Again, clarify the organ they analyzed. Intestine only? It is not good to exaggerate or obscure the description in the abstract.

Lines 32-34: "...through binding to enhancer regions, maintaining enhancer chromatin activity, and facilitating chromatin looping." Again, the results of intestine should not be universalized as concluding remarks.

Lines 36-58: From the perspective of impartiality, the main information known about HNF4, including the previous reports of this research group, should be included in Introduction.

Lines 49-51: "Most current brush border studies focus on genes encoding brush border structural proteins or cell biological characteristics of brush border assembly and function." Show refs.

Line 83: "within 50 kb..." Is it appropriate enough to detect the enhancer? Show refs describing rationale. As confirmed by Hi-C data, 75% enhancer-promoter interaction occurs within 500 Kbp from TSS (*Cell* 167, 1369-1384.e19 (2016)). The region within 200 Kbp from TSS contains only 29% in human and 37% in zebrafish (*Nucleic Acids Research* 48, 2357-2371, 2020). Therefore, the region within 50 Kb from TSS does not include the majority of enhancer information.

Line 86: The authors should perform DNase-seq of yolk sac for the comparison.

Lines 94-104: Fig. S1 should include yolk sac for the comparison.

Lines 105-124: Why don't you compare in between all 3 tissues in Fig.2?

Line 151-175: Given enhancer activation is often cell type-specific, kidney and yolk sac likely utilize unique and distinct enhancers to activate brush border gene program. Accordingly, chromatin looping could be different between these organs and intestine. I would like to know the active enhancer and chromatin looping in the kidney and yolk sac, in addition to the intestine.

Line 176: Using DKO mice, the same group previously reported that HNF4 is important for ISC maintenance (*Gastroenterology* 158, 985-999 (2020)) and intestinal epithelial identity (*Nature genetics* 51, 777-785 (2019)). Thus, in DKO mice, it is impossible to exclude the effects of ISC and enterocyte abnormalities and perform brush border-specific analysis. An experimental design that enables brush border-specific analysis is required.

Line 188: The abnormal intestinal morphology of DKO mice can be easily expected. In the situation where the same transcription factor HNF4 controls ISC maintenance and enterocyte identity, it is difficult to definitively conclude that HNF4 is required for brush border maintenance. Is there anything other than HNF4, such as the co-factor of HNF4, that is more brush border-specific?

Lines 212-218: Fig. 4E and 4G need quantitative data.

Line 219: The authors should test GSEA whether brush border genes are downregulated in yolk sac upon HNF4 loss and need to indicate functional impairment of yolk sac in HNF4 KO mice.

Line 256: The data that KO mice exhibit renal dysfunction due to brush border hypoplasia of the proximal tubule is very good. Functional verification is also required for other organs, i.e. nutrient absorption in the intestine, and fetal growth and nutrient absorption in the yolk sac.

Lines 268-273: Which transcription factors are abnormally activated and involved in the expression of the stress fiber genes by the HNF4 loss?

Lines 278-280: Which organ? Clarify the organ they analyzed. It is not good to exaggerate or obscure the description.

We thank the reviewers and editors for the favorable and helpful assessment of this work. We address each reviewer's critique in a point-by-point response below. The reviewers clearly made a significant effort in providing these suggestions, which we believe have resulted in a substantially improved manuscript. As the editors suggest, we show all changes in the manuscript file with yellow highlighting. We thank the reviewers for their efforts and hope that they find the resulting manuscript suitable for publication.

Reviewer #1 (Remarks to the Author):

How transcription factors are involved in programmes that generate sub-cellular structures is an interesting and important question. The brush border is an important cellular structure in tissues involved in fluid intake, such as the intestine or kidney. In this study, the authors provide compelling evidence that HNF4 is critical for the generation of the brush border in three different tissues, intestine, kidney and yolk sack. Thus, HNF4 seems to be the master regulator of brush border genesis. This work also reveals a potential model system for diseases such as Fanconi syndrome. The data are of high quality and the paper is very well presented. What is also really nice is the use of various complementary approaches to strengthen conclusions, such as ChIP-seq, HiCHIP, RNA-seq. Thus, this paper provides also interesting resources for the community.

We thank reviewer 1 for the favorable assessment of our work and its contribution to the field.

My only major comment is as follows:

While the study shows compellingly that HNF4 is essential for correct brush border formation in various tissues, it leaves open the exciting question: Is HNF4 sufficient to 'induce' brush border generation?

I think it would not be fair to ask to address this question in the mouse model, but I can imagine that one can devise a simple trans-(over)expression test in a cell culture system, e.g. using some fibroblast cell line that normally does not have a clear brush border. If such experiments do not provide evidence that HNF4 is sufficient to induce the program, that is ok and can be simply reported and figures shown as supplementary figures. However, if (over) expression of HNF4 causes the generation of some kind of brush border in a cell line that normally does not have it, this would be fantastic and increase the impact of this work further.

This is a great suggestion. We had the same idea when we were exploring the function of HNF4 in brush border. We found Chiba *et al.* (*J Cell Biol* 2006) already reported that overexpression of HNF4A could induce the formation of brush border in the F9 cells *in vitro*. We thus decided not to further pursue this idea in our study, and focused on brush border from different tissues *in vivo*. We thank reviewer for this idea, and we now include this important point in the Discussion of the revised manuscript (Page 8, line 324-325).

The authors could squeeze a bit more out of their rich data by performing some more bioinformatic analysis:

From their ChIP-seq data, to what extent are HNF4 binding sites found at putative enhancer

elements (more distal or downstream of genes) as opposed to promoter elements (please, note that H3K27ac is found at both enhancers and promoters)?

This is a great suggestion. We analyzed HNF4 binding sites with the *cis*-regulatory element annotation system (CEAS, v1.0.0), and we found most of the HNF4 binding sites are located at intron and distal intergenic regions (see figure below). As the reviewer suggests, we have included this analysis in Fig. S2a of the revised manuscript.

The motif analysis (Fig. 1g) highlights also CTCF/CTCFL which chimes with their finding of a role of HNF4 in having some role to gene loops. What is the relationship between HNF4 and CTCF binding sites at brush border genes?

This is an interesting idea. 3D-interactions within the genome are largely restricted to intra-chromosomal contacts, and within chromosomes, these interactions are largely partitioned into topologically associated domains (TADs). Each TAD typically includes hundreds of kb of chromatin and is anchored by the directional binding of CCCTC-binding factor (CTCF) to sequence-specific regions of the genome. CTCF complexes with cohesin proteins to stabilize these large chromosomal territories (Dixon et al., 2012; Phillips-Cremins et al., 2013; Rao et al., 2014; Vietri Rudan et al., 2015). By curating the public CTCF ChIP-seq (GSE49847), we think CTCF binding sites at brush border genes are more likely to function as architectural proteins, although we see a few overlapped binding sites of HNF4 and CTCF at brush border genes (see figure below). Because the number of overlapping regions between HNF4 and CTCF are low, we decided not to include this analysis in the revised manuscript, but can add it if the reviewer believes it will be helpful for our readers.

Fig. S9 should possibly move to the main figures, as this illustrates an interesting downstream effect of losing the brush border organisation. As supplementary figure, it is a bit hidden.

As the reviewer suggests, we have moved Fig. S9 to the main figures, and also added some new data to support this idea (see Fig. 6 and Fig. S12 in the revised manuscript).

Fig.3 C: Another effect of deletion of HNF4 seems to be loss of order of the intestinal epithelial cells. In WT, the cells are nicely aligned in rows, this is completely lost in the KO. Maybe this is something the authors should point out in their text, too.

As the reviewer suggests, we have included this point in the revised manuscript (Page 5, line 195-197).

Legend of Fig 1, (I) typo: 'intetsine' to 'intestine', please.

We apologize for this typo, and we correct it in the revised manuscript.

Reviewer #2 (Remarks to the Author):

Chen and colleagues here take a multi-omic approach to defining the biologic targets of HNF4 in brush border containing tissues intestine, yolk sac and kidney. Using RNA-seq and DNase-seq, they show that all three tissues commonly express most brush border genes,

and that HNF4 binding motifs are the most accessible of any transcription factor in the brush border geneset. ChIP-seq of HNF4A in kidney and intestine were 90% similar, suggesting a common gene regulation program across organs. Inducible knockout mice were generated and RNA-seq suggested a largely common set of downregulated genes in the KO enriched for brush border and transport genes and most of these are bound by HNF4A by ChIP-seq.

Using H3K27ac-MNase-ChIP-seq and H3K4me3 HiChIP, the authors provide evidence that KO of HNF closes chromatin and disrupts looping in brush border genes. They then characterize the morphologic and functional consequences of HNF4A KO across tissues. This is an interesting and technically well executed study. The strength lies in the multi-omic application of transcriptome and epigenome profiling techniques and linking these back to function, ie a common brush border gene regulatory program.

We thank reviewer 2 for highlighting the importance and interesting mechanisms identified by our work.

There are several things that would improve the manuscript:

1. While the multi-omic data is overall convincing, there is no validation of HNF4A targets by ChIP-seq in the KO tissues. Inclusion of validation data, even if simply select genes, would strengthen the work.

This is a good idea. As the reviewer suggests, we performed ChIP on intestinal WT and KO tissues under two different SDS conditions (0.1% and 0.125%). It seems the ChIP antibody is pretty good and our KOs are pretty clean. Unlike WT tissues (which showed a 10-fold to 40-fold enrichment at target sites in these experiments), we could not obtain enough DNA from KO tissues for further ChIP-qPCR (see figure below, n = 2 biological replicates per condition).

2. Fanconi syndrome is defined by glycosuria and amino aciduria as well as phosphate wasting. At a minimum, urine glucose should be measured though urinary phosphate would also be helpful.

This is a great suggestion. As the reviewer suggests, we measured urine glucose with a Glucose Assay Kit (ab65333, abcam), and we found increased glucose levels in the urine of mutants, which is consistent with Fanconi syndrome. We have included this data in Fig. 4j of the revised manuscript (Page 6, line 252-254; also see figure below).

3. The authors do point out that HNF4A has been knocked out in developing kidney but not in adult. It could be interesting to compare the magnitude of gene expression changes in Marable *et al.* with the results here. Presumably there would be fewer changes with the author's own dataset which helps to distinguish the utility of this dataset over the developmental one.

We thank the reviewer for this great idea. As the reviewer suggests, we re-analyzed the GSE112828 RNA-seq data of developing kidney (Marable *et al.*, *JCI insight* 2018), and compared this developing data set with our RNA-seq data of adult kidney. There are 286 genes are significantly altered in the developing kidney (FDR < 0.05, *Six2GFPcre*), and 265 of these genes (~93%) are also significantly altered in our RNA-seq data of adult kidney (FDR < 0.05, *UBC-Cre^{ERT2}*). We speculate the fewer significant genes we called from Marable *et al.* data set could be due to the mosaic expression of *Six2GFPcre*, but we do see a good overlap of significant genes between the developing kidney data and the adult kidney data upon HNF4A loss. We have included this important new data analysis in the revised manuscript (Page 9, line 350-352).

Minor:

1. It would be nice to see an analysis of stress fiber formation in KO kidney as well.

We thank reviewer for this nice idea, and we have included stress fiber staining of KO kidney vs WT in the revised manuscript (see Figure S12g & h, also see figure below). The keratin network (Keratin 8/18) plays a crucial role in stress fiber reinforcement (Fujiwara *et al.*, *Mol Biol Cell* 2016) and mechanotransduction (Cheah *et al.*, *PNAS* 2019). Additionally, calpains play an important role in focal adhesion and stress fiber formation (Glading *et al.*, *Trends in Cell Biology* 2002). Filamin A has two flexible hinge regions that are susceptible to proteolysis by calpain (Shao *et al.*, *Pathol Oncol Res* 2016). In the kidney, it seems that stress fibers are more enriched in distal tubules, as evidenced by staining of these stress fiber markers. For example, ABCG2 marks proximal tubules in the kidney, whereas Filamin A stains the distal tubules of kidney (see Figure S12h, also see figure below).

Unlike intestine and yolk sac (see new data in Figure 6a, b, e and Figure S12a-f, also see figures below), stress fibers are not stained in the cells with brush border (proximal tubules) in the kidney. We saw upregulated Keratin 18 and Calpain 2 in the kidney KO tissues. In healthy murine kidneys, Keratin 18 is not expressed in proximal or distal tubules and is only localized in the collecting ducts (Djudjaj *et al.*, *Kidney Int* 2016). Interestingly, Keratin 18 is observed *de novo* in distal tubules in kidney injury models (Djudjaj *et al.*,

Kidney Int 2016), and we found upregulated Keratin 18 in the kidney of *Hnf4α*^{KO} (Figure S12g), suggesting impaired kidney upon HNF4A loss. New data highlighting these findings and analyses are below, and also in the revised manuscript (Page 8, line 297-321).

Fig. S12

2. Can the authors present a fuller analysis of the genes with altered expression in the KO? Fig. 2C seems to selectively analyze only the brush border gene set.

The Fig. 2C related full table of GO term analysis is presented in Supplementary Data 5, providing a complete analysis of genes with altered expression in the KO.

Reviewer #3 (Remarks to the Author):

Chen et al. report a novel role of HNF4 in regulating expression of genes involved in formation of epithelial brush border in the intestine, kidney, and yolk sacs. Epithelial-specific silencing of *Hnf4* expression perturbed epithelial microvilli architecture in the intestine, kidney, and yolk sac while also promoting expression of genes involved in control of actin stress fibers and focal adhesions in the intestine. Mechanistically, it is proposed that HNF4 modulates the transcription of brush border genes by influencing the formation of chromatin loops when HNF4 is associated with enhancer chromatin structure, thereby resulting exposure of active promoter regions of these genes.

The overall role of HNF4 in the regulation of gene programs for microvilli and stress fibers formation in intestinal epithelial cells is novel and interesting. While most of the data

supports the proposed mechanisms, the study would be strengthened by a few additional experiments.

We thank reviewer 3 for the favorable assessment of our work and its novelty.

Minor Concerns:

1. In Fig. 2G, transcription of all genes analyzed (*Espn*, *Ezr*, *Myo7b*, and *Slc6a19*) is decreased in *Hnf4ag*DKO mice. Please include the expression of genes other than brush border genes as a negative control.

This is a good point, and we have included the transcript levels of non-brush border genes as controls in Fig. S4d of the revised manuscript (also see figure below, n = 3 biological replicates; N.S.: not significant).

2. In Fig. 3B, beta-actin expression in the apical compartment of epithelial cells should be quantified. Beta-actin expression seems to be reduced in immunofluorescence images (Fig.3B), but not in the western blotting results (Fig. 3D). Please elaborate on this discrepancy. To show F-actin organization, please consider presenting images after phalloidin staining and provide western blotting and imaging results showing other brush border proteins such as ezrin, villin and alkaline phosphatase.

As the reviewer suggests, we have quantified the beta-actin expression of Fig. 3B (current Fig. 3c in the revised manuscript), and the quantification has been included in Fig. S5b of the revised manuscript (also see figure below).

We think that while total protein of beta-actin was not changed in DKO vs WT, but the distribution of beta-actin in the cells was altered. Beta-actin was located at the brush border of intestinal villus cells of WT, but disrupted and redistributed throughout the cells upon HNF4 loss. We have included this point in the revised manuscript (Page 5, line 199-201).

We love the idea of phalloidin staining. As the reviewer suggests, we performed phalloidin staining (Thermo Fisher Scientific A34055) in the intestine of DKO vs WT. We found phalloidin staining showed similar intensity in the muscle layer of WT and DKO, but was specifically lost in the intestinal epithelial cells of DKO (see figure below). This finding is consistent with our previous staining of brush border markers, and we have included this new data in the revised manuscript (see revised Figure 2b).

3. The authors demonstrated expression of stress fibers/focal adhesions genes in the absence of HNF4. However, only upregulation of filamin protein expression in the intestine is shown. Please analyze expression of filamin and other focal adhesion proteins in the kidney and yolk sac epithelium of HNF4a KO mice.

We thank reviewer for this great idea, and we have included stress fiber/focal adhesion staining of intestine, kidney and yolk sac for WT vs KO in the revised manuscript (see new data in Figure 6b & e and Figure S12d, e, g, h, also see figure below).

To address this question, we purchased the following 7 antibodies related to stress fibers/focal adhesions, and 3 of these antibodies work well for IHC staining.

Antibodies that didn't work well in our hands: Vinculin Antibody (hVIN-1) NOVUS NB600-1293; Keratin 8/18 (C51) Mouse mAb Cell signaling #4546T; Anti-FHOD1 antibody (HPA024468-25UL Sigma-Aldrich); Vangl2 Antibody (C-2) Santa Cruz sc-515187.

Antibodies that worked well: Cytokeratin 18 Antibody (C-04) Santa Cruz sc-51582; Calpain 2 Antibody (E-10) Santa Cruz sc-373966; Laminin γ -2 Antibody (E-6) Santa Cruz sc-28330.

The keratin network (Keratin 8/18) plays a crucial role in stress fiber reinforcement (Fujiwara *et al.*, *Mol Biol Cell* 2016) and mechanotransduction (Cheah *et al.*, *PNAS* 2019). In healthy murine kidneys, Keratin 18 is not expressed in proximal or distal tubules and is only localized in the collecting ducts (Djudjaj *et al.*, *Kidney Int* 2016). Interestingly, Keratin 18 is observed *de novo* in distal tubules in kidney injury models (Djudjaj *et al.*, *Kidney Int* 2016). Additionally, calpains play an important role in focal adhesion and stress fiber formation (Glading *et al.*, *Trends in Cell Biology* 2002). Filamin A has two flexible hinge regions that are susceptible to proteolysis by calpain (Shao *et al.*, *Pathol Oncol Res* 2016). It seems that in kidney, stress fibers are more enriched in distal tubules, as evidenced by staining of these stress fiber markers. For example, ABCG2 marks the proximal tubules in the kidney,

whereas Filamin A stains the distal tubules of kidney (see Figure S12h, also see figure below). Unlike intestine and yolk sac, Filamin A is not stained in the cells with brush border (proximal tubules) in the kidney.

We found the transcript levels of Filamin A were upregulated in yolk sac upon Hnf4 loss (Figure S12f). We didn't see the elevated protein levels of Filamin A in the *Hnf4* α ^{KO} yolk sac tissues (Figure S12e), suggesting post-translational regulation or degradation may take place in Filamin A proteins.

Although we didn't see upregulated protein levels of Filamin A upon *Hnf4* loss in the kidney and yolk sac tissues, we found other stress fiber/focal adhesion markers, such as Keratin 18 and Calpain 2, were all upregulated upon Hnf4 loss in intestine (Figure 6e), kidney (Figure S12g) and yolk sac tissues (Figure 6b). Laminin γ -2, which functions in focal adhesion stability (Gagnoux-Palacios *et al.*, *J Biol Chem* 1996), was also upregulated upon *Hnf4* loss in intestine (Figure 6e) and yolk sac (Figure S12d) tissues.

Additionally, we also analyzed the stress fiber and focal adhesion related gene signatures from our new RNA-seq data of yolk sac for WT vs KO, and upregulated stress fiber/focal adhesion gene signatures were observed in yolk sac upon Hnf4 loss (Figure 6a and Figure S12a-c, f, also see figures below).

New data highlighting these findings and analyses are below, and also in the revised manuscript (Page 8, line 297-321).

Fig. S12

4. Please move Fig.S9 to the main figure as it provides important results related to stress fibers/focal adhesions in intestinal epithelial cells of *Hnf4a* DKO mice. Discuss how focal adhesion genes are upregulated in *Hnf4a* DKO mice. Since HNF4a has been reported to regulate intestinal epithelial barrier function, a discussion of the relationship between HNF4 and genes involved in controlling perijunctional actinomyosin ring and intercellular junctions will be useful.

As the reviewer suggests, we have moved Fig. S9 to the main figures, and also added some new data to support this idea (see Fig. 6 and Fig. S12 in the revised manuscript).

When cells are under mechanical stress, actin stress fibers increase to reinforce their mechanical strength. Elevated membrane tension lowers actin based-protrusion. Most stress fibers connect to focal adhesions and are important in mechanotransduction. We think upregulated stress fibers/focal adhesions are a consequence of impaired brush border upon HNF4 loss. We hypothesize that HNF4 transcription factors may work as a mechanosignaling sensor, and future study is needed to test this idea. We have added these points in the Discussion (Page 8, line 323-324 & 332-336).

5. In Fig.5H, yellow lines are difficult to see. Please change the color.

As the reviewer suggests, we have changed the color (see current Fig. 4h in the revised manuscript).

6. The word “intestine” is misspelled in line 535. Please modify.

We apologize for this typo, and we correct it in the revised manuscript.

Reviewer #4 (Remarks to the Author):

The brush border is a unique structure found in limited organs, e.g. intestine, kidney and yolk sac, and contributes to the absorption and transport of substances by dramatically increasing the surface area. This research group previously reported that transcription factor HNF4 is important for ISC maintenance (Gastroenterology. 2020 Mar; 158 (4): 985-999), and intestinal epithelial identity (Nature genetics 51, 777-785 (2019)). Here, the authors showed that HNF4 is also important for the maintenance of brush border in the intestine, kidney and yolk sac. HNF4 triggers the brush border gene program through binding to distal enhancer regions and facilitating chromatin looping at the gene loci of brush border genes in the intestine. This manuscript contains some interesting findings, but several major concerns are raised.

We thank reviewer 4 for their careful reading and thoughtful suggestions. We have addressed these suggestions and respond with changes as outlined in the point-by-point below.

Comments:

Line 21: “many” is not appropriate. Use “multiple” instead of “many”.

We appreciate the reviewer’s suggestion, and we made the suggested change in the revised manuscript.

Lines 25-26: “By ChIP-seq and RNA-seq, we find that HNF4 transcription factors bind to and activate the brush border gene program.” The authors should clearly indicate the organ they analyzed If they haven't analyzed all three organs

We appreciate this suggestion and have indicated the organs in the revised manuscript.

Lines 26-30: “To refine our understanding of HNF4-dependent regulation of brush border genes at the 3D chromatin architecture level, we apply H3K4me3 HiChIP-seq technology...” Again, clarify the organ they analyzed. Intestine only? It is not good to exaggerate or obscure the description in the abstract.

We are sorry about this, and we have clarified these details in the revised abstract.

Lines 32-34: “...through binding to enhancer regions, maintaining enhancer chromatin activity, and facilitating chromatin looping.” Again, the results of intestine should not be universalized as concluding remarks.

We are sorry about this again, and we have indicated the organ we tested in the revised abstract.

Lines 36-58: From the perspective of impartiality, the main information known about HNF4, including the previous reports of this research group, should be included in Introduction.

As the reviewer suggests, we have included our previous finding of HNF4 in the revised Introduction (Page 2, line 51-56).

Lines 49-51: “Most current brush border studies focus on genes encoding brush border structural proteins or cell biological characteristics of brush border assembly and function.” Show refs.

As the reviewer suggests, we have included more details and refs for the known studies in the revised Introduction (Page 2, line 45-49).

Line 83: “within 50 kb...” Is it appropriate enough to detect the enhancer? Show refs describing rationale. As confirmed by Hi-C data, 75% enhancer-promoter interaction occurs within 500 Kbp from TSS (Cell 167, 1369–1384.e19 (2016)). The region within 200 Kbp from TSS contains only 29% in human and 37% in zebrafish (Nucleic Acids Research 48, 2357–2371, 2020). Therefore, the region within 50 Kb from TSS does not include the majority of enhancer information.

This is a good question. In our previous experience, we found transcription factors such as CDX2 (Verzi *et al.*, *Molecular and Cellular Biology* 2011) and HNF4 (Chen *et al.*, *Nature Genetics* 2019) show a better correlation of their binding events and gene expression within 50 kb. For example, our intestine ChIP-seq shows ~70% of the HNF4 binding regions are within 50kb (see figure below, panel A, GREAT analysis). We further identified the nearby genes of HNF4 binding sites within 50 kb and 50-500 kb (panel B) and performed GSEA to compare the correlation of HNF4 binding with our RNA-seq data (DKO vs Control). The 898 genes within 50-500 kb of HNF4 binding sites do not show a good correlation with our RNA-seq data ($p = 0.2$, panel C), whereas genes within 50 kb of HNF4 binding sites show significant downregulation upon HNF4 loss ($p < 0.001$, panel C).

We also tried 500kb as a cutoff for identifying brush border gene enhancers as the reviewer suggests, and using these regions to identify transcription factor motifs, we found similar results as we saw in 50kb (See figure below).

Line 86: The authors should perform DNase-seq of yolk sac for the comparison.

When we initiated this project, we actually only worked on intestine and kidney. We curated public DNase-seq data of kidney and intestine (no available data of yolk sac in the database). Later on, we were encouraged by the strong phenotype of brush border in the intestine and kidney upon *Hnf4* loss, so we further extended our study to the brush border of yolk sac. While we have been able to generate RNA-seq and functional studies in the yolk sac, we could not complete DNase-seq at this time, but hope to investigate this question in the future.

Lines 94-104: Fig. S1 should include yolk sac for the comparison.

As the reviewer suggests, we have included the yolk sac in the revised Fig. S1.

Lines 105-124: Why don't you compare in between all 3 tissues in Fig.2?

In Fig. 2, we focus on the comparison of adult brush border tissues - intestine vs kidney. There is no HNF4 ChIP-seq data of yolk sac in the database, possibly due to the challenge of low materials of yolk sac epithelial tissues. Therefore, we took HNF4 ChIP-seq data of intestine from our lab, and we curated public HNF4 ChIP-seq data of kidney (GSE47192) when we initiated this project. Later, we were encouraged by the strong phenotypes of disrupted brush border in the intestine and kidney upon HNF4 loss, so we decided to keep exploring the phenotype of HNF4 loss in the yolk sac tissues.

Line 151-175: Given enhancer activation is often cell type-specific, kidney and yolk sac likely utilize unique and distinct enhancers to activate brush border gene program. Accordingly, chromatin looping could be different between these organs and intestine. I would like to know the active enhancer and chromatin looping in the kidney and yolk sac, in addition to the intestine.

This is a great point. We took advantage of available public ChIP-seq data of enhancer and promoter markers of intestine and kidney (GSE49847, Yue *et al.*, *Nature* 2014), and analyzed the enhancers at the brush border gene loci (see examples in the figure below). We found both similar and unique enhancers at the brush border gene loci of intestine and kidney. This is interesting, and we were planning to further compare the yolk sac ATAC-seq data with our intestine ATAC-seq and public kidney ATAC-seq data, as the first step to explore tissue-specific enhancers of brush border genes. It was pretty challenging to get a good cell prep of yolk sac epithelium. After many tries, we were able to send 3 ATAC-seq samples of yolk sac to GENEWIZ for sequencing. Unfortunately, unlike our other successful ATAC-seq, we encountered a problem for these yolk sac ATAC-seq samples. We recently got the data back, and it seems we got too many duplicates in these yolk sac ATAC-seq data, which we suspect was due to either poor cell prep quality or poor transposition. We are disappointed with the low quality of the yolk sac sequencing data, and we are sorry that we could not further explore this question. Due to the timeliness of this revision, we chose to begin the resubmission process now. We hope that the additional lines of evidence that link HNF4A to Yolk Sac brush border gene regulation that we presented in revision (RNAseq, co-staining, knockouts, and metabolomics) will be of sufficient interest to merit publication.

Line 176: Using DKO mice, the same group previously reported that HNF4 is important for ISC maintenance (Gastroenterology 158, 985-999 (2020)) and intestinal epithelial identity (Nature genetics 51, 777-785 (2019)). Thus, in DKO mice, it is impossible to exclude the effects of ISC and enterocyte abnormalities and perform brush border-specific analysis. An experimental design that enables brush border-specific analysis is required.

The brush border dysfunction is an important aspect of enterocyte abnormalities in our study. We agree that *Villin-Cre^{ERT2}* model could not exclude the effects of ISC. To focus on the enterocytes only, we are in the process of getting *Alpi-Cre^{ERT2}* mice. We have the same plan as the reviewer suggests, we will cross *Alpi-Cre^{ERT2}* mice with our DKO mice, but it will take probably 9 - 12 months to make a new model of *Hnf4af/f;Hnf4gcrispr/crispr;Alpi-Cre^{ERT2}* mice and their littermate controls. We hope we will have some exciting findings in this new mouse model in the future.

Line 188: The abnormal intestinal morphology of DKO mice can be easily expected. In the situation where the same transcription factor HNF4 controls ISC maintenance and

enterocyte identity, it is difficult to definitively conclude that HNF4 is required for brush border maintenance. Is there anything other than HNF4, such as the co-factor of HNF4, that is more brush border-specific?

This is a fascinating idea and one that we are actively pursuing. We performed RIME analysis to identify partner transcription factors of HNF4. This led us to 3 candidate partners. We are excited to test the hypothesis that one may be a brush border-specific gene.

Lines 212-218: Fig. 4E and 4G need quantitative data.

As the reviewer suggests, we quantified the staining of Fig 4E and 4G. As we adjusted the figure layout in the revised manuscript, now all these data could be found in Fig. S11 (also see figure below).

Line 219: The authors should test GSEA whether brush border genes are downregulated in yolk sac upon HNF4 loss and need to indicate functional impairment of yolk sac in HNF4 KO mice.

We thank reviewer for this very important point. As the reviewer suggests, we performed RNA-seq in yolk sac tissues of WT vs KO (4 replicates each), and we found brush border genes are also downregulated in yolk sac upon HNF4 loss (see Figure 5h in the revised manuscript, also see figure below).

In the revised manuscript, as the reviewer suggests, we also include new data of functional impairment analysis of yolk sac in HNF4 KO mice (see new data in revised Figure 5e-n and new Figure S10, also see figures below). We first tested the earliest time we could induce *Hnf4a* knockout in the pregnant female by gavaging tamoxifen & progesterone to prevent miscarriage (Figure 5e). We then measured the size and weight of WT and KO embryos, and we found reduced size and weight in *Hnf4α^{KO}* embryos compared to their *Cre*-negative

littermate controls (Figure 5f-g). By analyzing the new yolk sac RNA-seq data of WT vs KO, we found compromised brush border gene signatures upon HNF4A loss (Figure 5h & Figure S10a). As the brush border is defective in the yolk sac upon HNF4A loss, we wondered whether there was a problem of nutrient transport. To further address this question, we performed a metabolomics study by LC-MS for the amniotic fluid samples collected from WT and KO embryos. During pregnancy, amino acids represent one of the major nutrients for embryos. We found essential and non-essential amino acids (Figure 5i & Figure S10b) were all reduced in the amniotic fluid upon HNF4A loss. Coincidentally, amino acid transport genes are also compromised in the yolk sac of *Hnf4a*^{KO} (Figure 5j-k & Figure S10c), indicating defective nutrition transport upon HNF4A loss. Furthermore, compared to WT controls, we found increased uric acid in the amniotic fluid and compromised expression of uric acid transporters in the yolk sac of *Hnf4a*^{KO} (Figure 5l-n), suggesting waste removal might be also impaired upon HNF4A loss. All these findings suggest functional impairment of yolk sac in *Hnf4a*^{KO}. We have included all these new data in the revised manuscript (Page 7, line 266-288).

Fig. 5

Fig. S10

Line 256: The data that KO mice exhibit renal dysfunction due to brush border hypoplasia of the proximal tubule is very good. Functional verification is also required for other organs, i.e. nutrient absorption in the intestine, and fetal growth and nutrient absorption in the yolk sac.

We thank the reviewer for the nice words of our functional analysis in kidney KO. We actually published the intestinal function-related data in our previous studies, including the rapid body weight loss and a distended and fluid-filled intestine in the *Hnf4α*^{DKO} (see figure

below, first reference). By using Shh-Cre to investigate the function of HNF4 in the developing intestine, we found Hnf4 α DKO cannot survive after birth and loss of 3 Hnf4 alleles in the developing gut resulted in growth retardation after birth (see information below on panels to view, second reference), suggesting the important function of HNF4 in nutrient absorption in the intestine. We have included these previous findings in the revised manuscript (Page 5, line 181-182).

[Redacted]

See:

Fig. 1d and Supplementary Fig. 4a (Nat Genet. 2019 May;51(5):777-785. doi: 10.1038/s41588-019-0384-0.)

And:

**Supplementary Fig. 3B and Supplementary Fig. 3C (Development. 2019 Aug 6;146(19):dev179432. doi: 10.12/d
ev.179432.)**

Regarding to the nutrient absorption in the yolk sac, please find our new data in the previous point above, and Page 7, line 266-288.

Lines 268-273: Which transcription factors are abnormally activated and involved in the expression of the stress fiber genes by the HNF4 loss?

This is a great idea. To pursue this idea, we performed HOMER motif analysis on the accessible chromatin regions of the stress fiber genes (TSS +/- 50 kb) that are significantly altered upon HNF4 loss in the intestine. Due to the limited chromatin regions associated with these activated genes (total only 240 accessible chromatin regions), the p values of the HOMER analysis were not as striking as we usually see in other cases (see figure below, left

panel), and this might compromise the power of motif prediction. Among the transcription factors we identified, we found KLF TFs are abnormally activated upon HNF4 loss in the intestine, including *Klf5*, *Klf7*, *Klf10*, *Klf11* and *Klf13* (see figure below, right panel, RNA-seq n=3). While this is interesting analysis, we are not convinced it is rigorous enough to include in the present manuscript, unless the reviewer would advise us otherwise.

Lines 278-280: Which organ? Clarify the organ they analyzed. It is not good to exaggerate or obscure the description.

We are sorry about this again, and we have indicated the organ we tested in the revised manuscript.

Reviewers' Comments:

Reviewer #2:

Remarks to the Author:

The authors have added important new data in many cases and in others have made good faith attempts to do so, and the text has been clarified and updated. This is an excellent contribution.

Reviewer #3:

Remarks to the Author:

The authors have thoughtfully addressed our concerns.

Reviewer #4:

Remarks to the Author:

Regarding yolk sac, which lacked data, the authors conducted additional experiments and gave generally satisfactory answers to some of my comments, although some experiments did not work. One thing I'm not satisfied with is that it remains unclear whether the phenotype seen in the intestinal brush border in Villin-CreERT2; Hnf4af/f; Hnf4gCrispr/Crispr mouse model (DKO mice) is indirect due to abnormal differentiation of ISC, or the HNF4 deficiency of mature enterocytes directly causes the brush border phenotype. In particular, the same group previously reported that HNF4 is important for ISC maintenance (Gastroenterology 158, 985-999 (2020)), meaning the authors notice the DKO mice are unable to address my concern described above. Indeed, they agree in their replies, "this model could not exclude the effects of ISC" and they are planning to generate Alpi-CreERT2; Hnf4af/f; Hnf4gCrispr/Crispr mice, in which the HNF4 in ISC is intact. To prove the direct effects of HNF4 deficiency of mature enterocytes on the brush border phenotype, the analysis using this new model is required.

We thank the reviewers and editors for the favorable and helpful assessment of this work. We address each reviewer's critique in a point-by-point response below.

Reviewer #2 (Remarks to the Author):

The authors have added important new data in many cases and in others have made good faith attempts to do so, and the text has been clarified and updated. This is an excellent contribution.

We thank reviewer 2 for the favorable assessment of our work.

Reviewer #3 (Remarks to the Author):

The authors have thoughtfully addressed our concerns.

We thank reviewer 3 for the favorable assessment of our work.

Reviewer #4 (Remarks to the Author):

Regarding yolk sac, which lacked data, the authors conducted additional experiments and gave generally satisfactory answers to some of my comments, although some experiments did not work. One thing I'm not satisfied with is that it remains unclear whether the phenotype seen in the intestinal brush border in Villin-CreERT2; Hnf4af/f; Hnf4gCrispr/Crispr mouse model (DKO mice) is indirect due to abnormal differentiation of ISC, or the HNF4 deficiency of mature enterocytes directly causes the brush border phenotype. In particular, the same group previously reported that HNF4 is important for ISC maintenance (Gastroenterology 158, 985-999 (2020)), meaning the authors notice the DKO mice are unable to address my concern described above. Indeed, they agree in their replies, "this model could not exclude the effects of ISC" and they are planning to generate Alpi-CreERT2; Hnf4af/f; Hnf4gCrispr/Crispr mice, in which the HNF4 in ISC is intact. To prove the direct effects of HNF4 deficiency of mature enterocytes on the brush border phenotype, the analysis using this new model is required.

We thank reviewer 4 for their thoughtful suggestions. We have discussed this point as a limitation of the current work in the revised manuscript.